# History Aware Multimodal Transformer for Vision-and-Language Navigation

**Shizhe Chen, Pierre-Louis Guhur, Cordelia Schmid, Ivan Laptev**

Inria, École normale supérieure, CNRS, PSL Research University

{shizhe.chen, pierre-louis.guhur, cordelia.schmid, ivan.laptev}@inria.fr

https://cshizhe.github.io/projects/vln_hamt.html

## Abstract

Vision-and-language navigation (VLN) aims to build autonomous visual agents that follow instructions and navigate in real scenes. To remember previously visited locations and actions taken, most approaches to VLN implement memory using recurrent states. Instead, we introduce a History Aware Multimodal Transformer (HAMT) to incorporate a long-horizon history into multimodal decision making. HAMT efficiently encodes all the past panoramic observations via a hierarchical vision transformer (ViT), which first encodes individual images with ViT, then models spatial relation between images in a panoramic observation and finally takes into account temporal relation between panoramas in the history. It, then, jointly combines text, history and current observation to predict the next action. We first train HAMT end-to-end using several proxy tasks including single step action prediction and spatial relation prediction, and then use reinforcement learning to further improve the navigation policy. HAMT achieves new state of the art on a broad range of VLN tasks, including VLN with *fine-grained instructions* (R2R, RxR), *high-level instructions* (R2R-Last, REVERIE), *dialogs* (CVDN) as well as *long-horizon VLN* (R4R, R2R-Back). We demonstrate HAMT to be particularly effective for navigation tasks with longer trajectories.

## 1 Introduction

Vision-and-language navigation (VLN) has recently received growing attention [1, 2, 3, 4, 5]. VLN requires an agent to understand natural language instructions, perceive the visual world, and perform navigation actions to arrive at a target location. A number of datasets have been proposed to support various VLN tasks such as indoor and outdoor navigation with fine-grained instructions [2, 6, 7], language-driven remote object finding [8] and navigation in dialogs [9].

VLN agents are faced with several challenges. First, as opposed to static vision-text grounding [10], the agent continuously receives new visual observations and should align them with instructions. Most of existing works adopt recurrent neural networks (RNNs) [6, 11, 12, 13, 14, 15, 16] to encode historical observations and actions within a fixed-size state vector to predict the next action. Such condensed states might be sub-optimal for capturing essential information in extended trajectories [17]. For instance, *"bring the spoon to me"* requires the agent to remember its start location after navigating to the *"spoon"*, while early memories are prone to fade in the recurrent state. Few endeavors [18, 19] construct external map-like memories for received observations. Nevertheless, these approaches still rely on RNNs to track the navigation state. As the history plays an important role in environment understanding and instruction grounding, we propose to explicitly encode the history as a sequence of previous actions and observations instead of using recurrent states.

35th Conference on Neural Information Processing Systems (NeurIPS 2021).

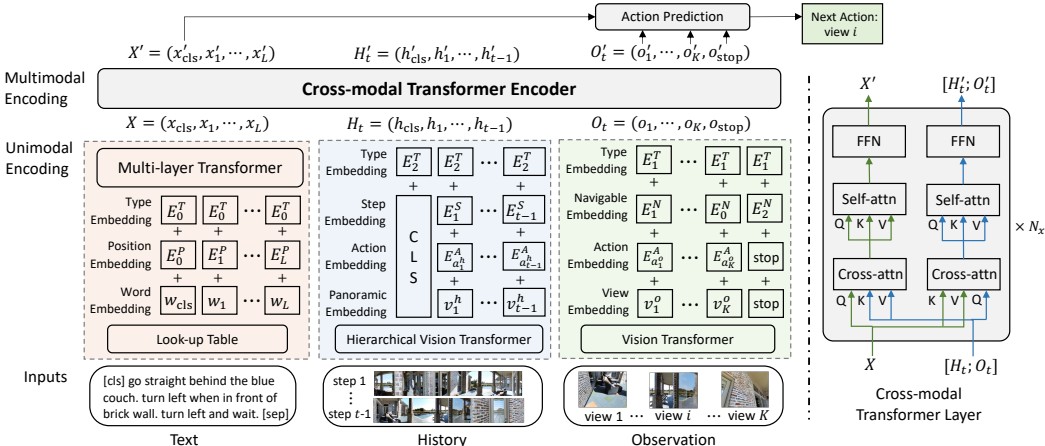

Figure 1: The architecture of History Aware Multimodal Tranformer (HAMT). HAMT jointly encodes textual instruction, full history of previous observations and actions, and current observation to predict the next action.

Another VLN challenge concerns the generalizations of agents to new environments that have not been observed during training [4]. One direction is to learn more generic text-image representations. The PRESS model [20] improves language representation with a pretrained BERT encoder [21], and PREVALENT [22] uses pairs of instruction and single-step observations to pretrain a multimodal transformer. Though achieved promising results, these works do not optimize visual representation for the target navigation task. Moreover, lack of history in training [22] makes it hard to learn cross-modal alignment and increases the risk of overfitting to training environments. Another direction towards better generalization is to overcome exposure bias [23] due to discrepancy between training and inference. Different methods have been adopted for VLN including DAgger [6, 24] and scheduled sampling [20, 25]. Reinforcement Learning (RL) [12, 26] is one of the most effective approach among them, but it is considered unstable to directly train large-scale transformers via RL [27].

To address the above challenges, we propose the History Aware Multimodal Transformer (HAMT), a fully transformer-based architecture for multimodal decision making in VLN tasks. As illustrated in Figure 1, HAMT consists of unimodal transformers for text, history and observation encoding, and a cross-modal transformer to capture long-range dependencies of the history sequence, current observation and instruction. Since our history contains a sequence of all previous observations, its encoding is computationally expensive. To resolve complexity issues, we propose a hierarchical vision transformer as shown in Figure 2, which progressively learns representations for a single view, spatial relationships among views within a panorama and, finally, the temporal dynamics across panoramas of the history. In order to learn better visual representations, we propose auxiliary proxy tasks for end-to-end training. Such tasks include single-step action prediction based on imitation learning, self-supervised spatial relationship reasoning, masked language and image predictions and instruction-trajectory matching. We empirically show that our training facilitates the subsequent fine-tuning of our model with RL [28]. We carry out extensive experiments on various VLN tasks, including VLN with *fine-grained instructions* (R2R [6] and RxR [7]), *high-level instructions* (REVERIE [8] and our proposed R2R-Last), *dialogs* [9] as well as *long-horizon VLN* (R4R [3] and our proposed R2R-Back which requires the agent to return back after arriving at the target location). HAMT outperforms state of the art on both seen and unseen environments in all the tasks.

We summarize our contributions as follows: (1) We introduce HAMT to efficiently model long-horizon history of observed panoramas and actions via hierarchical vision transformer; (2) We train HAMT with auxiliary proxy tasks in an end-to-end fashion and use RL to improve the navigation policy; (3) We validate our method and outperform state of the art in a diverse range of VLN tasks, while demonstrating larger gains for long-horizon navigation.

## 2   Related work

**Vision-and-language navigation.** Training instruction-following navigation agents has attracted increasing research attention [1, 2, 6, 7, 8, 29]. Anderson *et al.* [6] propose a sequence-to-sequence

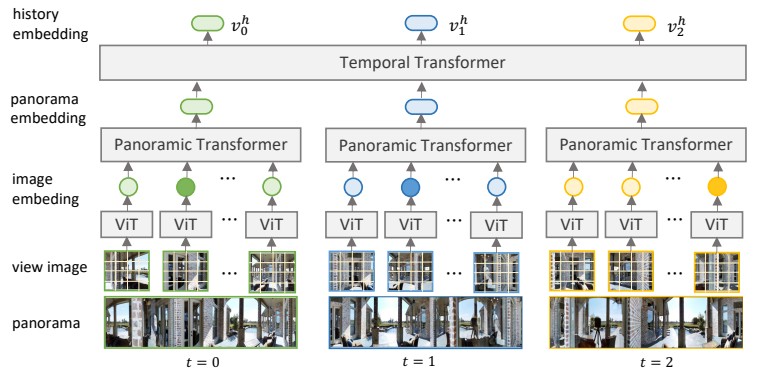

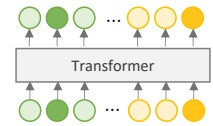

(b) **Flattened history encoding**. It encodes spatial and temporal relations at the same time.

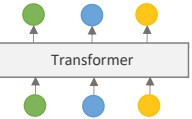

(a) **Hierarchical history encoding**. It first encodes individual view images with ViT, then models the spatial relation between images in each panorama, and finally captures the temporal relation between panoramas in the history.

(c) **Temporal-only history encoding**. It only considers temporal relation of oriented views.

Figure 2: A comparison of history encoding methods. Circle nodes in different colors denote view images of panorama at different steps. Darker circle nodes are the oriented view of the agent.

LSTM baseline for the VLN task. Fried *et al.* [11] extend it with panoramic action space and synthesized instructions. To improve cross-modal alignment, the self-monitoring agent [13] proposes co-grounding and progress estimation, and RelGraph [15] uses graphs to model relationships across scene, objects and directions. Reinforcement learning (RL) is typically used to improve navigation policy. The EnvDrop model [12] mixes imitation learning and A3C [28]. The RCM [14] utilizes intrinsic reward of cross-modal matching in REINFORCE algorithm. Wang *et al.* [30] propose to learn rewards via soft expert distillation. Due to the success of transformer [31], recent works explore transformer architectures in VLN. PRESS [20] replaces LSTM instruction encoder with pretrained BERT [21]. SIA [16] uses transformer for single-step multimodal fusion and LSTM for sequential action prediction. PTA [32] is a transformer VLN model using CNNs to extract visual features [33]. Here we propose the first full transformer architecture for VLN and train it end-to-end.

**Memory-based policy for navigation.** LSTMs [34] have been the dominant approach to encode memories for navigation [6, 11, 12, 14]. Condensing all history into one feature vector, however, is prone to the loss of information. Alternative approaches include topological map memory structures [35, 36]. Deng *et al.* [18] use graphs to capture environment layout and enable long-term planing. A similar graph is adopted in [19] with frontier-exploration based decision making. But these works still utilize LSTMs for state tracking. To exploit long-term spatio-temporal dependencies, Fang *et al.* [17] store histories in a sequence encoded with transformer. Recurrent VLN-BERT [5] injects a recurrent unit to encode histories in transformer for VLN. The most similar work to ours is Episodic Transformer (E.T.) [37]. Differently from [37], we propose a hierarchical encoding of the panoramic observation history and optimize the whole model in end-to-end training.

**Multimodal pretraining with transformers.** Recent works show significant progress in vision and language tasks using multimodal pretraining. In particular, transformer architectures such as one-stream [38, 39] and dual-stream [40, 41] achieve state of the art for a number of downstream tasks including visual question answering, image-text retrieval and image captioning. While most previous methods rely on CNN to extract image representations, ViLT [42] adopts Vision Transformer (ViT) [43] and trains it with associated texts in an end-to-end manner thanks to the efficiency of ViT. A few endeavors [22, 44] explore multimodal pretraining for VLN. PREVALENT [22] pretrains a transformer using instructions and single-step observations without referring to trajectory history. VLN-BERT [44] measures the compatibility between an instruction and images in a path but does not support action prediction. Our work presents the first end-to-end trainable VLN transformer that jointly encodes text, history and observation, and is able to sequentially predict actions.

## 3 Method

**Problem definition** The VLN problem [6] is formulated as a partially observable Markov decision process, where future observations are independent of the past conditioning on current state $s_t$. Given

an instruction $\mathcal{W}$ containing a sequence of $L$ words $(w_1, w_2, \cdots, w_L)$, an agent should follow the instruction to move in a connectivity graph to reach the goal location. At each step $t$, the agent receives an observation $\mathcal{O}_t$, a panorama of its surrounding environment. The $\mathcal{O}_t$ consists of $K$ single view images split from the panorama $\mathcal{O}_t \triangleq ([v_1^o; a_1^o], \cdots, [v_K^o; a_K^o])$, where $v_i^o$ is the visual feature of the $i$-th view and $a_i^o$ denotes the relative angle to face the view (subscript $t$ is omitted for simplicity). There are $n$ navigable viewpoints among all the $K$ views[1], denoted as $\mathcal{O}_t^c \triangleq ([v_1^c; a_1^c], \cdots, [v_n^c; a_n^c])$. We follow the setup in [11] and use $\mathcal{O}_t^c$ as the decision space, so the agent only needs to select a candidate in $\mathcal{O}_t^c$ at each step. All observations $\mathcal{O}_i$ and performed actions $a_i^h$ before step $t$ form the history $\mathcal{H}_t \triangleq ([\mathcal{O}_1; a_1^h], \cdots, [\mathcal{O}_{t-1}; a_{t-1}^h])$, where $a_i^h$ denotes the turned angles at step $i$. The goal is to learn a policy $\pi$ parametrized by $\Theta$ to predict the next action based on the instruction, history and the current observation, which is $\pi(a_t | \mathcal{W}, \mathcal{H}_t, \mathcal{O}_t, \mathcal{O}_t^c; \Theta)$.

Unlike dominant recurrent approaches to condense $\mathcal{H}_t$ into a fixed-size vector, in this section, we present the History Aware Multimodal Transformer (HAMT) that jointly encodes text, long-horizon history, and observation for sequential action prediction. The model architecture is described in Section 3.1. We propose end-to-end training for HAMT in Section 3.2 to learn unimodal and multimodal representations, and then use RL to fine-tune the navigation policy in Section 3.3.

### 3.1 HAMT: History Aware Multimodal Transformer

Figure 1 illustrates the model architecture of HAMT. The inputs text $\mathcal{W}$, history $\mathcal{H}_t$ and observation $\mathcal{O}_t$ are first encoded via the corresponding unimodal transformers respectively, and then fed into the cross-modal transformer encoder to capture multimodal relationships.

**Text Encoding.** For each token $i$ in the instruction $\mathcal{W}$, we embed it as the summation of its word embedding $w_i$, position embedding $E_i^P$ and type embedding of text $E_0^T$. Then we employ a transformer with $N_L$ layers to obtain contextual representation $x_i$ following the standard BERT [21].

**Observation Encoding.** For each view $[v_i^o; a_i^o]$ in the panoramic observation $\mathcal{O}_t$, we first represent the relative angle $a_i^o$ as $E_{a_i^o}^A = (\sin \theta_i, \cos \theta_i, \sin \phi_i, \cos \phi_i)$ where $\theta_i$ and $\phi_i$ are the relative heading and elevation angle to the agent's orientation. Then the observation embedding $o_i$ is as follows:

$$o_i = \text{LN}(W_v^o v_i^o) + \text{LN}(W_a^o E_{a_i^o}^A) + E_{o_i}^N + E_1^T \tag{1}$$

where $W_v^o, W_a^o$ are learnable weights. The $E_{o_i}^N$ denotes the navigable embedding to differentiate types of views, with $E_0^N$ for non-navigable view, $E_1^N$ for navigable view and $E_2^N$ for stop view (we append a stop token in observation to support stop action). The $E_1^T$ is the type embedding of observation. We omit bias terms for simplicity. The LN denotes layer normalization [45]. Because $a_i^o$ has much lower feature dimensions than $v_i^o$, we apply LN to balance the encoded $a_i^o$ and $v_i^o$.

**Hierarchical History Encoding.** As $\mathcal{H}_t$ consists of all the past panoramic observations $\mathcal{O}_i$ and performed actions $a_i^h$ before step $t$, it is important to encode $\mathcal{H}_t$ efficiently as context. Figures 2b-2c depict the flattened and temporal-only history encoding approaches used in VLN-BERT [44] and E.T. [37] respectively. The flattened approach treats each view image in $\mathcal{O}_i$ as a token, so the history sequence contains $tK$ tokens. Though it enables to learn relationships among all image views, the computation cost quadratically increases with the sequence length, making it inefficient for long-horizon tasks. In the temporal-only approach, only the oriented view of the agent in each $\mathcal{O}_i$ is taken as inputs instead of the whole panorama, so only $t$ temporal tokens are encoded. However, this approach can lose critical information in past observations. For example, in the instruction *"with the windows on your left, walk through the large room past the sitting areas"*, the object *"window"* does not appear in the oriented view of the agent. Therefore, the encoded history is insufficient to tell whether the agent passed the window or not, making the model confused to take the next action.

In order to balance computational efficiency and information integrity, we propose a hierarchical history encoding approach as illustrated in Figure 2a. It hierarchically encodes view images within each panorama and then temporal relationships across panoramas, similar to the factorized spatial-temporal video transformer [46]. For each $\mathcal{O}_i$, its constituent view images are first embedded via ViT and Eq (1), and then encoded via a panoramic transformer with $N_h$ layers to learn spatial relationships within the panorama. We apply average pooling to obtain panorama embedding, and add it with the

---

[1]A navigable view can lead to one or multiple viewpoints. We follow [5, 12] to use different features for these viewpoints. The viewpoints share the same visual features but differ in angle features.

Table 1: Comparison of HAMT and previous VLN transformers.

| Models | Inputs | | | Proxy Tasks | | | | |
|---|---|---|---|---|---|---|---|---|
| | Text | History | Observation | MLM | MRM | ITM | SAP/SAR | SPREL |
| PREVALENT [22] | ✓ | | ✓ | ✓ | | | ✓ | |
| VLN-BERT [44] | ✓ | ✓ | | ✓ | ✓ | ✓ | | |
| HAMT (Ours) | ✓ | ✓ | ✓ | ✓ | ✓ | ✓ | ✓ | ✓ |

oriented view image feature in residual connection. The parameters in ViT and panoramic transformer are shared for different steps. In this way, each historical observation $\mathcal{O}_i$ is represented as $v_i^h$, and the final temporal token $h_i$ is computed as:

$$h_i = \text{LN}(W_v^h v_i^h) + \text{LN}(W_a^h E_{a_i^h}^A) + E_i^S + E_2^T \tag{2}$$

where $E_i^S$ denotes the $i$-th step embedding, $E_2^T$ is the type embedding of history. The computational cost is $O(tK^2 + t^2)$, which significantly reduces from $O(t^2 K^2)$ in the flattened approach. To be noted, we add a special token [cls] to the start of the history sequence to obtain a global representation. The embedding of [cls] is a parameter to learn, which is initialized from a zero vector.

**Cross-modal Encoding.** We concatenate history and observation as the vision modality, and use cross-modal transformer with $N_x$ layers to fuse features from text, history and observation as shown in the right of Figure 1. The reason of using such dual-stream architecture rather than one-stream is that the length of different modalities can be highly imbalanced, and the dual-stream architecture can balance the importance of intra- and inter-modal relationships by model design [47]. In each cross-modal layer, a vision-text cross-attention is firstly performed for vision modality to attend relevant text information and vice versa for text modality. Then each modality uses self-attention to learn intra-modal relationship such as interaction between observation and history, followed by a fully-connected neural network. Finally, the HAMT model outputs embeddings $X' = (x'_{\text{cls}}, x'_1, \cdots, x'_L), H'_t = (h'_{\text{cls}}, h'_1, \cdots, h'_{t-1}), O'_t = (o'_1, \cdots, o'_K, o'_{\text{stop}})$ for tokens in text, history and observation respectively.

## 3.2 End-to-end training with proxy tasks

As it is difficult to train large-scale transformers with RL due to sparse supervision [27], we propose to first end-to-end train HAMT via several proxy tasks to learn unimodal and multimodal representation.

Table 1 compares our HAMT with previous VLN transformers PREVALENT [22] and VLN-BERT [44] in inputs and proxy tasks. As neither PREVALENT nor VLN-BERT jointly encodes text, history and observation, a limited choice of proxy tasks can be applied in training. Our model instead can take advantage of various proxy tasks to learn cross-modal alignment, spatial and temporal reasoning, and history-aware action prediction. Given the input pair $(\mathcal{W}, \mathcal{H}_T)$ where $T$ is the length of full trajectory, we can apply common proxy tasks as in vision-and-language pretraining [40, 44], including Masked Language Modeling (MLM), Masked Region Modeling (MRM) and Instruction Trajectory Matching (ITM). Details of the three proxy tasks are presented in the supplementary material. In the following, we introduce new proxy tasks given the triplet input $(\mathcal{W}, \mathcal{H}_t, \mathcal{O}_t)$ specifically for VLN tasks.

**Single-step Action Prediction/Regression (SAP/SAR).** The task deploys imitation learning to predict the next action based on instruction, history from expert demonstration and the current observation. We formulate it as a classification and a regression task respectively. In the SAP classification task, we predict action probability for each navigable view in $\mathcal{O}_t^c$ which is $p_t(o'_i) = \frac{\exp(f_{\text{SAP}}(o'_i \odot x'_{\text{cls}}))}{\sum_j \exp(f_{\text{SAP}}(o'_j \odot x'_{\text{cls}}))}$, where $f_{\text{SAP}}$ is a two-layer fully-connected network, $\odot$ is element-wise multiplication and $x'_{\text{cls}}$ is output embedding of special text token [cls]. The objective is to minimize negative log probability of the target view action $o'_*$: $L_{\text{SAP}} = -\log p_t(o'_*)$. In SAR regression task, we directly predict the action heading and elevation angles based on the text token [cls] which is $\hat{\theta}_t, \hat{\phi}_t = f_{\text{SAR}}(x'_{\text{cls}})$. The loss function is $L_{\text{SAR}} = (\hat{\theta}_t - \theta_t)^2 + (\hat{\phi}_t - \phi_t)^2$. The two proxy tasks enable the model to learn how to make action decision conditioning on instruction and contextual history.

**Spatial Relationship Prediction (SPREL).** Expressions of egocentric and allocentric spatial relations are frequent in navigational instructions, such as *"walk into the room on your left"* and *"enter*

*the bedroom next to the stairs"*. In order to learn spatial relation aware representations, we propose the SPREL self-supervised task to predict relative spatial position of two views in a panorama based on only visual feature, angle feature or both. Assume $[v_i^o; a_i^o]$ and $[v_j^o; a_j^o]$ are two views in $\mathcal{O}_t$, we randomly zero out $v_*^o$ or $a_*^o$ with probability of 0.3. Their encoded representations are $o_i'$ and $o_j'$, and their relative heading and elevation angles are $\theta_{ij}, \phi_{ij}$. We then predict $\hat{\theta}_{ij}, \hat{\phi}_{ij} = f_{\text{SPREL}}([o_i'; o_j'])$ where $[;]$ denotes vector concatenation and optimize $L_{\text{SPREL}} = (\hat{\theta}_{ij} - \theta_{ij})^2 + (\hat{\phi}_{ij} - \phi_{ij})^2$. The task helps for spatial relationship reasoning in the observation.

**Training Strategy.** Instead of directly training the whole HAMT model at once, we propose to progressively train HAMT in two stages. In the first stage, we freeze ViT pretrained on ImageNet [48] and train the rest of the modules which are randomly initialized. This aims to avoid catastrophic forgetting of the pretrained weights in ViT. Then we unfreeze ViT and train the whole model end-to-end. The learning rate for ViT is set to be higher than for others modules to avoid vanishing gradients and to speedup convergence. We empirically show that the proposed two-stage training outperforms one-stage training in the supplementary material.

### 3.3 Fine-tuning for sequential action prediction

**Structure Variants.** We present two variants of HAMT for action prediction in the following. 1) MLP action head: we directly reuse the action prediction network $f_{\text{SAP}}$ in the SAP task to predict navigable views. We use it as default for VLN tasks. 2) MLP action head based on encoder-decoder structure: the original HAMT model applies cross-modal attention for both vision-to-text and text-to-vision, which is computationally expensive when instructions are long. Therefore, we remove the cross-modal attention from text to vision. In this way, we separate the cross-modal transformer into an encoder which only takes instruction as input, and a decoder that inputs history and observation as query and attends over encoded text tokens. Please see supplementary material for details.

**RL+IL Objective.** We combine Reinforcement Learning (RL) and Imitation Learning (IL) to fine-tune HAMT for sequential action prediction. The IL relies on the SAP loss defined in Section 3.2 and follows the expert action at each step while RL samples actions according to the policy $\pi$. Specifically, we use the Asynchronous Advantage Actor-Critic (A3C) RL algorithm [28]. At each step $t$, the agent samples an action based on policy $\pi$: $\hat{a}_t^h \sim \pi(a_t | \mathcal{W}, \mathcal{H}_t, \mathcal{O}_t, \mathcal{O}_t^c)$ and receives an immediate reward $r_t$. For non-stop actions, we set $r_t$ as the reduced distance of taking the action to the target and the increased alignment score [3] compared to expert demonstration as defined in [5]; for the stop action, $r_t = 2$ if the agent successfully arrives at the target otherwise -2. A critic network is trained to estimate the value of each state $s_t$, which is $R_t = \sum_{k=0}^{T-t} \gamma^k r_{t+k}$ where $\gamma$ is discount factor. We implement it as $V_t = f_{\text{critic}}(x_{\text{cls}}' \odot h_{\text{cls}}')$. As the reward signal favors shortest distance, we empirically find it benefits to combine A3C RL with IL weighted by $\lambda$, which is:

$$\Theta \leftarrow \Theta + \mu \frac{1}{T} \underbrace{\sum_{t=1}^{T} \nabla_\Theta \log \pi(\hat{a}_t^h; \Theta)(R_t - V_t)}_{\text{Reinforcement Learning (RL)}} + \lambda \mu \frac{1}{T^*} \underbrace{\sum_{t=1}^{T^*} \nabla_\Theta \log \pi(a_t^*; \Theta)}_{\text{Imitation Learning (IL)}} \tag{3}$$

where $\mu$ is the learning rate, $a_t^*$ is the expert action at step $t$ of the expert trajectory of length $T^*$.

## 4 Experiments

### 4.1 Experimental setup

**Datasets.** We evaluate our method on four VLN tasks (seven datasets): *VLN with fine-grained instructions* (R2R [6], RxR [7]); *VLN with high-level instructions* (REVERIE [8], R2R-Last); *vision-and-dialogue navigation* (CVDN [9]); and *long-horizon VLN* (R4R [3], R2R-Back).

- **R2R** [1] builds upon Matterport3D [49] and includes 90 photo-realistic houses with 10,567 panoramas. It contains 7,189 shortest-path trajectories, each associated with 3 instructions. The dataset is split into train, val seen, val unseen and test unseen sets with 61, 56, 11 and 18 houses respectively. Houses in val seen split are the same as training, while houses in val unseen and test splits are different from training.

- **RxR** [7] is a large multilingual VLN dataset based on Matterport 3D. The instructions are in three different languages (English, Hindi and Telugu). The dataset emphasizes the role of language in VLN by addressing biases in paths and describing more visible entities than R2R.
- **R4R** [3] extends R2R dataset by concatenating two adjacent tail-to-head trajectories in R2R. Therefore, it has longer instructions and trajectories. The trajectories are also less biased as they are not necessarily the shortest-path from start to end location.
- **R2R-Back** is a new VLN setup proposed in this work. The agent is required to return to its start location after arriving at the destination. The agent needs to remember its navigation histories to solve the task. We add a return command at the end of each instruction in R2R and a reverse path from the end to start locations as expert demonstration.
- **CVDN** [9] defines a navigation from dialog history task, which requires an agent to arrive at goal regions based on multi-turn question-answering dialogs. Such types of instructions are often ambiguous and under-specified. The lengths of instructions and paths are also long.
- **REVERIE** [8] replaces step-by-step instructions in R2R with high-level instructions, which mainly describe the target location and object. The agent, hence, is required to navigate to the goal without detailed guidance and depends on its past experiences.
- **R2R-Last** is our proposed VLN setup similar to REVERIE. It only uses the last sentence from the original R2R instructions describing the final destination.

**Evaluation metrics.** We adopt standard metrics [1], including (1) Trajectory Length (TL): the agent's navigated path in meters; (2) Navigation Error (NE): the average distance in meters between the agent's final position and the target; (3) Success Rate (SR): the ratio of trajectories reaching the destination with a maximum error of 3 meters to the target; and (4) Success Rate normalized by the ratio between the length of the shortest path and the predicted path (SPL). SPL is more relevant than SR as it balances the navigation accuracy and efficiency. For long-horizon VLN task (R4R and R2R-Back), we further employ three metrics to measure the path fidelity between the predicted path and target path, including (5) Coverage weighted by Length Score (CLS) [3]; (6) the normalized Dynamic Time Warping (nDTW) [50]; and (7) the Success weighted by nDTW (SDTW).

**Implementation details.** For the HAMT model, we set $N_L = 9$ for language transformer, $N_h = 2$ for panoramic transformer in hierarchical history encoding, and $N_x = 4$ for cross-modal transformer. There are $K = 36$ view images in each panoramic observation. We use ViT-B/16 [43] for image encoding if not otherwise specified. In training with proxy tasks, we randomly select proxy tasks for each mini-batch with predefined ratio. We train HAMT for 200k iterations with fixed ViT using learning rate of 5e-5 and batch size of 64 on 4 NVIDIA Tesla P100 GPUs ($\sim$1 day). The whole HAMT model is trained end-to-end for 20k iterations on 20 NVIDIA V100 GPUs with learning rate of 5e-5 for ViT and 1e-5 for the others ($\sim$20 hours). We use R2R training set and augmented pairs from [22] for training unless otherwise noted. In fine-tuning with RL+IL, we set $\lambda = 0.2$ in Eq (3) and $\gamma = 0.9$. The model is fine-tuned for 100k iterations with learning rate of 1e-5 and batch size of 8 on a single GPU. Unimodal encoders are fixed by default. The best model is selected according to performance on val unseen split. We use the same augmented data as [5] for R2R for fair comparison, while no augmented data is used for other datasets. Greedy search is applied in inference following the single-run setting. Please see supplementary material for more details.

## 4.2 Ablation studies

In this section, we evaluate each component in the HAMT model, including: hierarchical history encoding, end-to-end training with proxy tasks, and fine-tuning objectives.

**How important is the history encoding for VLN?** For fair comparison with the state-of-the-art recurrent architecture RecBERT [5], we use the same Resnet152 visual features and train all the models from scratch with RL+IL objectives to avoid the influence of different weight initialization. The models are optimized for 300k iterations end-to-end except for the visual feature. Table 2 compares different history encoding approaches on

Table 2: R2R navigation results for alternative methods of history encoding. All methods use Resnet152 visual features and are trained from scratch on R2R dataset.

| History | Val Seen | | Val Unseen | |
| Encoding | SR↑ | SPL↑ | SR↑ | SPL↑ |
| --- | --- | --- | --- | --- |
| RecBERT [5] | 62 | 59 | 50 | 46 |
| Recurrent | $60.9_{\pm 1.0}$ | $56.6_{\pm 1.1}$ | $52.2_{\pm 0.7}$ | $47.0_{\pm 0.5}$ |
| Temporal-only | $61.5_{\pm 0.8}$ | $57.7_{\pm 0.7}$ | $53.2_{\pm 0.1}$ | $48.0_{\pm 0.4}$ |
| Hierarchical | $\mathbf{65.5}_{\pm 1.2}$ | $\mathbf{61.3}_{\pm 1.4}$ | $\mathbf{54.4}_{\pm 0.4}$ | $\mathbf{48.7}_{\pm 0.4}$ |

R2R dataset. Our recurrent model slightly differs from RecBERT (no init. OSCAR) [5] in trans-

Table 3: Ablations for end-to-end HAMT training on R2R dataset using proposed proxy tasks.

(a) Comparison of visual features and end-to-end training. The "PT" stands for proxy tasks in training; "e2e" for optimizing the visual representation.

| feature | PT | e2e | Val Seen | | Val Unseen | |
|---|---|---|---|---|---|---|
| | | | SR↑ | SPL↑ | SR↑ | SPL↑ |
| Resnet 152 | × | × | $65.5_{\pm1.2}$ | $61.3_{\pm1.4}$ | $54.4_{\pm0.4}$ | $48.7_{\pm0.4}$ |
| | ✓ | × | $69.3_{\pm1.0}$ | $64.8_{\pm1.2}$ | $63.5_{\pm0.5}$ | $57.5_{\pm0.5}$ |
| ViT | ✓ | × | $\mathbf{75.7}_{\pm1.0}$ | $\mathbf{72.5}_{\pm1.0}$ | $64.4_{\pm0.3}$ | $58.8_{\pm0.0}$ |
| | ✓ | ✓ | $75.0_{\pm0.9}$ | $71.7_{\pm0.7}$ | $\mathbf{65.7}_{\pm0.7}$ | $\mathbf{60.9}_{\pm0.7}$ |

(b) Comparison of different proxy tasks. The "SAP(R)" denotes the single step action prediction and regression task, and "SPREL" is the spatial relationship prediction task.

| SAP (R) | SP REL | Val Seen | | Val Unseen | |
|---|---|---|---|---|---|
| | | SR↑ | SPL↑ | SR↑ | SPL↑ |
| × | × | $71.2_{\pm2.3}$ | $67.2_{\pm2.0}$ | $62.8_{\pm1.3}$ | $57.7_{\pm1.0}$ |
| ✓ | × | $74.7_{\pm0.6}$ | $71.1_{\pm0.9}$ | $63.6_{\pm0.1}$ | $58.1_{\pm0.4}$ |
| ✓ | ✓ | $\mathbf{75.7}_{\pm1.0}$ | $\mathbf{72.5}_{\pm1.0}$ | $\mathbf{64.4}_{\pm0.3}$ | $\mathbf{58.8}_{\pm0.0}$ |

former architecture as shown in Figure 1. It achieves slightly better performance on val unseen split. The temporal-only model uses transformer to encode agent's oriented visual observations in history sequence, and outperforms the recurrent method by relative gains of 1.9% on SR and 2.1% on SPL for val unseen split. Adding panoramic observations in a hierarchical way results in 4.2% (SR) and 3.6% (SLP) relative improvements on the val unseen split compared to the recurrent method. Even larger improvements are achieved on val seen split as the hierarchical model has a larger capacity to fit the seen environments. This evaluation demonstrates the advantage of our hierarchical history representation compared to the recurrent and temporal-only history representation.

**How much does training with proxy tasks help?** We next evaluate the advantage of training HAMT end-to-end with proxy tasks. In Table 3a, the first row uses RL+IL objectives to train HAMT from scratch, while the second row uses proxy tasks for training prior to RL+IL fine-tuning. We can see that it significantly boosts the performance to first train with proxy tasks. It improves on val unseen split with 16.7% and 18.0% relative gains on SR and SPL respectively, indicating that training with auxiliary proxy tasks enables better generalization. In the third row, we replace the visual feature from Resnet152 to ViT. The ViT feature improves the performance on both val seen and val unseen splits, showing that more powerful visual representations matter. Finally, training ViT end-to-end obtains 2.1% gains on SPL on val unseen split. This is the first time to show that optimizing visual representations end-to-end is beneficial for VLN tasks. In Table 3b, we evaluate the benefit of the two new proxy tasks for frozen ViT features using the other proxy tasks by default. The SAP(R) uses imitation learning to predict actions, which directly influences the navigation policy and improves the performance by a large margin. The SPREL is a self-supervised proxy task that forces the model to learn spatial relationships in panorama and helps generalization in unseen environments. More experiments to ablate contributions from history encoding and proxy tasks, contributions of proxy tasks in end-to-end training *etc*. are presented in supplementary material.

**What is the impact of the fine-tuning objectives?** Table 4 presents results using different objectives in fine-tuning. The first row directly applies HAMT trained by proxy tasks, which achieves lower performance than that after IL fine-tuning, because we mainly use augmented data in proxy task training to increase visual diversity, but such noisy data deteriorates action prediction performance. Previous work [12] has shown that RL alone performs poorly. However, training with

Table 4: Ablations for fine-tuning objectives of sequential action prediction on R2R dataset.

| IL | RL | Val Seen | | Val Unseen | |
|---|---|---|---|---|---|
| | | SR↑ | SPL↑ | SR↑ | SPL↑ |
| × | × | 57.9 | 54.8 | 51.8 | 48.9 |
| ✓ | × | $63.7_{\pm2.1}$ | $61.7_{\pm2.2}$ | $57.2_{\pm0.1}$ | $54.7_{\pm0.3}$ |
| × | ✓ | $70.5_{\pm2.9}$ | $65.6_{\pm2.8}$ | $63.5_{\pm1.4}$ | $57.5_{\pm1.1}$ |
| ✓ | ✓ | $\mathbf{75.0}_{\pm0.9}$ | $\mathbf{71.7}_{\pm0.7}$ | $\mathbf{65.7}_{\pm0.7}$ | $\mathbf{60.9}_{\pm0.7}$ |

proxy tasks stabilizes the followup RL fine-tuning. HAMT optimized by RL achieves much better performance than that when fine-tuning with IL on the SR metric. It indicates that RL is able to learn better exploration strategy on unseen environments. However, as the reward for RL focuses more on shortest paths rather than path fidelity with instructions, the improvement on SPL metric is relatively small compared to SR metric. Moreover, the fluctuation of the pure RL objective is larger than IL. Therefore, mixing the RL and IL achieves the best performance.

### 4.3 Comparison to state of the art

**VLN with fine-grained instructions: R2R and RxR.** Table 5 compares HAMT with previous VLN methods on the R2R benchmark. Our model outperforms state-of-the-art results of RecBERT [5]

Table 5: Comparison with state-of-the-art methods on R2R dataset.

| Methods | Validation Seen | | | | Validation Unseen | | | | Test Unseen | | | |
|---|---|---|---|---|---|---|---|---|---|---|---|---|
| | TL | NE↓ | SR↑ | SPL↑ | TL | NE↓ | SR↑ | SPL↑ | TL | NE↓ | SR↑ | SPL↑ |
| Seq2Seq [6] | 11.33 | 6.01 | 39 | - | 8.39 | 7.81 | 22 | - | 8.13 | 7.85 | 20 | 18 |
| SF [11] | - | 3.36 | 66 | - | - | 6.62 | 35 | - | 14.82 | 6.62 | 35 | 28 |
| PRESS [20] | 10.57 | 4.39 | 58 | 55 | 10.36 | 5.28 | 49 | 45 | 10.77 | 5.49 | 49 | 45 |
| EnvDrop [12] | 11.00 | 3.99 | 62 | 59 | 10.70 | 5.22 | 52 | 48 | 11.66 | 5.23 | 51 | 47 |
| AuxRN [51] | - | 3.33 | 70 | 67 | - | 5.28 | 55 | 50 | - | 5.15 | 55 | 51 |
| PREVALENT [22] | 10.32 | 3.67 | 69 | 65 | 10.19 | 4.71 | 58 | 53 | 10.51 | 5.30 | 54 | 51 |
| RelGraph [15] | 10.13 | 3.47 | 67 | 65 | 9.99 | 4.73 | 57 | 53 | 10.29 | 4.75 | 55 | 52 |
| RecBERT [5] | 11.13 | 2.90 | 72 | 68 | 12.01 | 3.93 | 63 | 57 | 12.35 | 4.09 | 63 | 57 |
| HAMT (Ours) | 11.15 | **2.51** | **76** | **72** | 11.46 | **2.29** | **66** | **61** | 12.27 | **3.93** | **65** | **60** |

by relative 5.9% and 7.0% improvements in SPL on val seen and unseen splits respectively. We achieve state-of-the-art performance under the single-run setting on the unseen testing split of the leaderboard[2]. It demonstrates the effectiveness and generalization of our model. We further provide computation time in inference for HAMT and RecBERT in the supplementary material to show the efficiency of our HAMT model. We also achieve large improvements on RxR dataset. The full results are presented in supplementary material.

**Long-horizon VLN: R4R and R2R-Back.** Table 6 shows navigation results on R4R dataset. As R4R contains longer instructions and trajectories compared to R2R, we use the encoder-decoder variant of HAMT for better efficiency. Our method outperforms previous approaches in all metrics and shows particularly large improvements for the path fidelity related metrics. Compared to RecBERT, HAMT achives 8.2% and 9.5% relative improvement in CLS and nDTW respectively. The large improvements on these path fidelity related metrics indicate that HAMT is better to follow the designated path of the fine-grained instruction. Figure 3 evaluates the performance of HAMT and RecBERT with respect to instruction length measured by words. Though the nDTW decreases for longer instructions, the relative improvement of HAMT increases with the instruction length.

Table 6: Comparison on R4R val unseen split.

| Methods | NE↓ | SR↑ | CLS↑ | nDTW↑ | SDTW↑ |
|---|---|---|---|---|---|
| SF [11] | 8.47 | 24 | 30 | - | - |
| RCM [14] | - | 29 | 35 | 30 | 13 |
| PTA [32] | 8.25 | 24 | 37 | 32 | 10 |
| EGP [18] | 8.0 | 30.2 | 44.4 | 37.4 | 17.5 |
| RelGraph [15] | 7.43 | 36 | 41 | 47 | 34 |
| RecBERT[†] [5] | 6.67 | 43.6 | 51.4 | 45.1 | 29.9 |
| HAMT (Ours) | **6.09** | **44.6** | **57.7** | **50.3** | **31.8** |

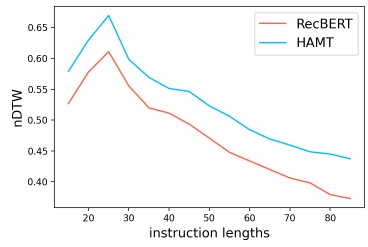

Figure 3: nDTW with respect to instruction length on R4R val unseen split.

The navigation performance on R2R-Back dataset is presented in Table 7. We compare with two state-of-the-art recurrent models EnvDrop [12] and RecBERT [5] based on LSTM and transformer respectively (both models are trained on R2R-Back for fair comparison). The improvements are more significant on this task as it requires the agent to remember the way it came to the target to successfully return back. The recurrent state is insufficient to capture such history and leads to inferior performance compared to the HAMT model.

Table 7: Comparison of methods on the R2R-Back dataset.

| Methods | Val Seen | | | | | Val Unseen | | | | |
|---|---|---|---|---|---|---|---|---|---|---|
| | TL | SR↑ | SPL↑ | nDTW↑ | SDTW↑ | TL | SR↑ | SPL↑ | nDTW↑ | SDTW↑ |
| EnvDrop[†] [12] | 23.83 | 44.1 | 42.0 | 61.3 | 39.4 | 24.57 | 32.4 | 30.2 | 51.1 | 28.0 |
| RecBERT[†] [5] | 22.33 | 51.4 | 48.4 | 67.3 | 45.7 | 23.35 | 41.1 | 37.7 | 58.2 | 35.6 |
| HAMT (Ours) | 22.76 | **64.8** | **61.8** | **73.7** | **58.9** | 23.78 | **57.2** | **53.1** | **65.1** | **49.5** |

---

[2]We report the published results on the testing unseen split as shown in `https://eval.ai/web/challenges/challenge-page/97/leaderboard/270` (25/10/2021).

Table 8: Navigation performance on CVDN dataset.

|  | Val Seen | Val Unseen | Test Unseen |
|---|---|---|---|
| PREVALENT [22] | - | 3.15 | 2.44 |
| VISITRON [52] | 5.11 | 3.25 | 3.11 |
| MT-RCM+EnvAg [53] | 5.07 | 4.65 | 3.91 |
| HAMT (Ours) | **6.91** | **5.13** | **5.58** |

**Vision-and-Dialog Navigation: CVDN.**
The CVDN dataset contains dialogs as instructions and use Goal Progress (GP) in meters as the primary evaluation metric. GP measures the difference between completed distance and left distance to the goal, so the higher the better. There are two types of demonstrations in the dataset. One is shortest-path trajectory and the other is player's navigation trajectory. We mix the two types of demonstrations as supervision in training which has shown to be the most effective in previous works [22, 52, 53]. As navigation paths in CVDN dataset are much longer than R2R dataset, we adopt the encoder-decoder variant of HAMT. As shown in Table 8, HAMT outperforms existing recurrent approaches on both seen and unseen environments, and achieves the top position in the leaderboard[3]. It demonstrates that our HAMT model is generalizable to different types of instructions in new VLN tasks.

**VLN with high-level instructions: R2R-Last and REVERIE.** Table 9 shows results on the R2R-Last dataset that specifies the goal location and contains no step-by-step instructions. The HAMT model with the hierarchical history encoding is able to better accumulate the knowledge of the environment and achieves 9.8% and 10.5% relative gains on SPL metric on seen and unseen splits respectively compared to RecBERT [5]. The REVERIE dataset also

Table 9: Comparison on the R2R-Last dataset.

| Methods | Val Seen | | Val Unseen | |
|---|---|---|---|---|
|  | SR↑ | SPL↑ | SR↑ | SPL↑ |
| EnvDrop[†] [12] | 42.8 | 38.4 | 34.3 | 28.3 |
| RecBERT[†] [5] | 50.2 | 45.8 | 41.6 | 37.3 |
| HAMT (Ours) | **53.3** | **50.3** | **45.2** | **41.2** |

contains high-level instructions but requires object grounding at the target location besides navigation. We provide results on REVERIE dataset in supplementary material. Our HAMT achieves SPL 30.20 and 26.67 on val unseen and test splits respectively, outperforming the state of the art navigation performance [5] by 5.3% and 2.7%.

## 5 Conclusion

This paper presents the first end-to-end transformer for vision-and-language navigation, denoted as History Aware Multimodal Transformer (HAMT). Our method efficiently encodes long-horizon history and combines it with instructions and observations to derive multimodal action prediction. The HAMT is first trained with proxy tasks in an end-to-end manner, and is then fine-tuned with RL to improve the navigation policy. We achieve state-of-the-art navigation performance on a diverse range of challenging VLN tasks, demonstrating improved accuracy and generalization of our approach compared to the dominant recurrent methods. Future work could extend our history-aware transformer to VLN with continuous actions [54] and could benefit from pretraining on larger navigation datasets. This paper has minimal ethical, privacy and safety concerns.

## Acknowledgments and Disclosure of Funding

This work was granted access to the HPC resources of IDRIS under the allocation 101002 made by GENCI. It was funded in part by the French government under management of Agence Nationale de la Recherche as part of the "Investissements d'avenir" program, reference ANR19-P3IA-0001 (PRAIRIE 3IA Institute) and by Louis Vuitton ENS Chair on Artificial Intelligence.

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
