# History Aware Multimodal Transformer for Vision-and-Language Navigation
# – Supplementary Material –

**Shizhe Chen, Pierre-Louis Guhur, Cordelia Schmid, Ivan Laptev**

Inria, École normale supérieure, CNRS, PSL Research University

{shizhe.chen, pierre-louis.guhur, cordelia.schmid, ivan.laptev}@inria.fr

https://cshizhe.github.io/projects/vln_hamt.html

Section A provides additional details for the model. The experimental setup is described in Section B, including datasets, metrics and implementation details. Section C presents computation time on R2R dataset and full experimental results on RxR and REVERIE datasets. Section D includes more ablations. Finally, Section E illustrates qualitative results.

## A  Model details

### A.1  Proxy tasks in training

We employ five proxy tasks to train HAMT and introduced SAP/SAR and SPREL in Section 3.2. In the following, we present the other three proxy tasks, which are all based on the input pair $(\mathcal{W}, \mathcal{H}_T)$, where $\mathcal{W}$ is the textual instruction and $\mathcal{H}_T$ is the full trajectory with length $T$.

**Masked Language Modeling (MLM).** The task predicts masked words based on contextual words and the full trajectory. We randomly mask out tokens in $\mathcal{W}$ with the probability of 15% with a special token [mask] as in BERT, and predict the word distribution $p(w_i|\mathcal{W}_{\setminus i}, \mathcal{H}_T) = f_{\mathrm{MLM}}(x'_i)$ where $\mathcal{W}_{\setminus i}$ is the masked instruction, $x'_i$ is the output embedding of the masked word $w_i$ and $f_{\mathrm{MLM}}$ is a two-layer fully-connected network. The objective is to minimize the negative log-likelihood of original words: $L_{\mathrm{MLM}} = -\log p(w_i|\mathcal{W}_{\setminus i}, \mathcal{H}_T)$. The task is beneficial to learn grounded language representations and cross-modal alignment.

**Masked Region Modeling (MRM).** The task aims to predict semantic labels of masked observations in the trajectory given an instruction and neighboring observations. We zero out observations in $\mathcal{H}_T$ 15% of the time. The target of a masked $\mathcal{O}_i$ is the class probability predicted by an image classification model pretrained on ImageNet. We use ViT-B/16 [1] in this work. Suppose $P_i \in \mathbb{R}^{1000}$ is the target class probability for a masked $\mathcal{O}_i$, we predict $\hat{P}_i = f_{\mathrm{MRM}}(o'_i)$ where $o'_i$ is the output embedding of masked $\mathcal{O}_i$, and minimize the KL divergence between the two probability distributions: $L_{\mathrm{MRM}} = -\sum_{j=1}^{1000} P_{i,j} \log \hat{P}_{i,j}$. In order to solve the task, $o'_i$ should capture temporal continuity in the history sequence and align with relevant instructions.

**Instruction Trajectory Matching (ITM).** The task predicts whether a pair of instruction and trajectory is aligned. We predict the alignment score as $s(\mathcal{W}, \mathcal{H}_T) = f_{\mathrm{ITM}}(x'_{\mathrm{cls}} \odot h'_{\mathrm{cls}})$, where $\odot$ is element-wise multiplication and $x'_{\mathrm{cls}}, h'_{\mathrm{cls}}$ are output embeddings for the text [cls] token and the history [cls] token respectively. We sample 4 negative trajectories for each positive instruction-trajectory pair during training, in which two negative trajectories are randomly selected from other positive pairs in the mini-batch, two are obtained by temporally shuffling the positive trajectory. The objective is the Noisy Contrastive Estimation loss [2]: $L_{\mathrm{ITM}} = -\log \frac{\exp(s(\mathcal{W}, \mathcal{H}_T))}{\exp(s(\mathcal{W}, \mathcal{H}_T)) + \sum_{k=1}^{4} \exp(s(\mathcal{W}, \mathcal{H}_{T,k}^{\mathrm{neg}}))}$. The model is supposed to learn cross-modal alignment and be sensitive to temporal orders of history to solve the task.

35th Conference on Neural Information Processing Systems (NeurIPS 2021).

## A.2 Structure variants in fine-tuning

We present the encoder-decoder variant of HAMT in fine-tuning on the right of Figure 1. Compared to the original cross-modal transformer on the left, the variant removes text-to-vision cross-modal attention. The encoder encodes the texts to obtain textual embeddings. Then the decoder reuses the same text embeddings in vision-to-text attention layer at each navigation step. In this way, the variant is more efficient when instructions are long *e.g.* in R4R and RxR datasets.

Figure 1: Comparison of the original cross-modal transformer layer (left) and the encoder-decoder based variant (right).

# B Experimental setup

## B.1 Dataset details

Table 1 summarizes details of the dataset split. The proposed R2R-Back and R2R-Last setups consider exactly the same splits as the R2R dataset. We present details to construct R2R-Back and R2R-Last in the following.

Table 1: Dataset statistics. #traj, #instr denote the number of trajectories and instructions respectively.

| Dataset | Train | | Val Seen | | Val Unseen | | Test Unseen | |
|---|---|---|---|---|---|---|---|---|
| | #traj | #instr | #traj | #instr | #traj | #instr | #traj | #instr |
| R2R [3] | 4,675 | 14,039 | 340 | 1,021 | 783 | 2,349 | 1,391 | 4,173 |
| RxR [4] | 11,077 | 79,467 | 1,244 | 8,813 | 1,517 | 13,652 | - | 11,888 |
| R4R [5] | 25,921 | 233,532 | 115 | 1,035 | 5,026 | 45,234 | - | - |
| R2R-Back | 4,675 | 14,039 | 340 | 1,021 | 783 | 2,349 | - | - |
| CVDN [6] | 4,742 | 4,742 | 382 | 382 | 907 | 907 | 1,384 | 1,384 |
| R2R-Last | 4,675 | 14,039 | 340 | 1,021 | 783 | 2,349 | - | - |
| REVERIE [7] | 4,150 | 10,466 | 515 | 1,423 | 1,328 | 3,521 | 2,304 | 6,292 |

**R2R-Back.** We append a returning command at the end of annotated instructions in R2R to create new instructions for R2R-Back. The returning command is randomly sampled from the following sentences: "walk back to the start", "return by the way you came", "double back to where you start", "backtrack to the start", "back the way you came", "return to the starting point". The original target location is viewed as a middle stop point. The groundtruth trajectory in R2R-Back is the concatenation of the original and its inverse trajectory.

**R2R-Last.** We use spacy toolkit[1] to split sentences for instructions in R2R. We only select the last sentence in each instruction as the new high-level instruction. It mainly describes where the goal location is *e.g.* "stop in front of the vent", requiring the agent to explore houses without step-by-step textual guidance. The groundtruth trajectory is the same as R2R.

## B.2 Evaluation Metrics

In R2R, RxR, R4R and R2R-Last datasets, a predicted trajectory is considered to be successful if the agent arrives 3 meters near to the final destination. However, such definition would make a motionless agent achieve 100% success rate (SR) on R2R-Back dataset as the final destination is the same as the starting location. Therefore, in R2R-Back evaluation, we define the success as that an agent firstly arrives 3 meters near to the original destination and then returns 3 meters near to its starting location. The groundtruth length in the SPL metric is also modified as the total traversed distance in groundtruth trajectory rather than the shortest distance between start and target location. As the REVERIE task aims for remote object grounding, the success on REVERIE is defined as arriving at a viewpoint where the target object is visible.

---

[1] https://spacy.io/

## B.3 Implementation Details

**Training with proxy tasks.** We sample proxy tasks for each mini-batch to train the HAMT model. The sampling ratio is MLM:MRM:ITM:SAP:SAR:SPREL=5:2:2:1:1:1. The optimizer is AdamW [8]. In the end-to-end training stage, we use image augmentation and regularization techniques to avoid overfitting of the ViT model, including RandAugment [9] and stochastic depth [10].

**Fine-tuning for sequential action prediction.** Due to different goals in various VLN tasks, we design different rewards in reinforcement learning for each downstream VLN dataset. In R2R, RxR and R4R datasets, the reward is introduced in Section 3.3 to take both goal distance and path fidelity into account. In R2R-Last, REVERIE and CVDN datasets where the instruction may not describe detailed navigation path, we only use the reduced distance to the goal viewpoints as rewards. We normalize the reduced distance in the same way as in the R2R dataset. In R2R-Back dataset, we use a different fine-tune strategy to avoid trivial motionless solutions. We require the agent to predict stop actions twice for the original destination (midpoint) and its starting point (final destination) respectively. Before arriving at the midpoint, the RL reward is computed based on distances to the midpoint. If the agent predicts a wrong location to stop for the midpoint, the episode is stopped; otherwise the agent continues its task while receiving rewards based on the distance to the final destination for fine-tuning. We run each experiment twice for ablation study and use the best result on the validation unseen split for the state-of-the-art comparison.

## C Experimental results

### C.1 Computation Efficiency

To assess the influence of history encoding on the inference time, we compare HAMT with RecBERT [11]. The HAMT and RecBERT use the same number of layers in the language transformer and cross-modal transformer. The main difference of two models is in the history encoding and the attended length of history for action prediction. We run each model on the R2R val unseen split (2349 instructions) and report inference times averaged over two runs using a single Tesla P100 GPU. For our method we

Table 2: Computation time in inference on R2R val unseen split.

|  | Inference Time (s) | SR | SPL |
|---|---|---|---|
| RecBERT [11] | 69 | 63 | 57 |
| HAMT | 104 | 66 | 61 |
| HAMT noT2V | 76 | 65 | 60 |

compare variants with and without Text-to-Vision Attention (see Section A.2), denoted here as HAMT and HAMT noT2V respectively. We can see that HAMT and its noT2V variant are only 1.5x and 1.1x slower compared to RecBERT, suggesting that attending to the whole history does not increase the inference time significantly. Moreover, while HAMT noT2V is only 10% slower compared to [11], it still outperforms [11] in SR and SPL on val unseen split.

### C.2 RxR dataset

As shown in Table 1, RxR dataset contains much more instructions than R2R dataset. Therefore, we directly use RxR in training proxy tasks rather than R2R with augmented data. As there are three different languages in RxR, we take advantage of pretrained multilingual BERT [12] to initialize the unimodal language encoder, so we are able to deal with multilingual instructions using the same HAMT

Table 3: Navigation performance on RxR test split.

|  | PL | SR↑ | SPL↑ | nDTW↑ | SDTW↑ |
|---|---|---|---|---|---|
| Multilingual Baseline [4] | 16.88 | 20.98 | 18.55 | 41.05 | 20.59 |
| Monolingual Baseline [4] | 17.05 | 25.40 | 22.59 | 41.05 | 20.59 |
| CLIP-ViL | 15.43 | 38.34 | 35.17 | 51.10 | 32.42 |
| CLEAR-CLIP | 16.46 | 40.29 | 36.57 | 53.69 | 34.86 |
| Multilingual HAMT | 19.77 | **53.12** | **46.62** | **59.94** | **45.19** |
| Human | 20.78 | 93.92 | 74.13 | 79.48 | 76.90 |

model. We employ the encoder-decoder variant of HAMT for computational efficiency. For fair comparison with other approaches in RxR testing leaderboard[2] which adopt pretrained CLIP [13] features, we use the same visual features without end-to-end optimization. Table 3 presents navigation performances on RxR test split. Our multilingual HAMT model achieves 12.83% and 6.25% gains on

---

[2]https://ai.google.com/research/rxr/competition?active_tab=leaderboard (25/10/2021).

Table 4: Navigation performances on RxR val seen and val unseen splits.

| | Val Seen | | | | Val Unseen | | | |
|---|---|---|---|---|---|---|---|---|
| | SR↑ | SPL↑ | nDTW↑ | SDTW↑ | SR↑ | SPL↑ | nDTW↑ | SDTW↑ |
| Multilingual Baseline [4] | 25.2 | - | 42.2 | 20.7 | 22.8 | - | 38.9 | 18.2 |
| Monolingual Baseline [4] | 28.8 | - | 46.8 | 23.8 | 28.5 | - | 44.5 | 23.1 |
| Multilingual HAMT | **59.4** | **58.9** | **65.3** | **50.9** | **56.5** | **56.0** | **63.1** | **48.3** |

Table 5: Navigation and object grounding performances on REVERIE val unseen and test splits.

| | Validation Unseen | | | | | | Test Unseen | | | | |
|---|---|---|---|---|---|---|---|---|---|---|---|
| Methods | | Navigation | | | Grounding | | | Navigation | | | Grounding | |
| | TL | SR↑ | OSR↑ | SPL↑ | RGS↑ | RGSPL↑ | TL | SR↑ | OSR↑ | SPL↑ | RGS↑ | RGSPL↑ |
| Seq2Seq [3] | 11.07 | 4.20 | 8.07 | 2.84 | 2.16 | 1.63 | 10.89 | 3.99 | 6.88 | 3.09 | 2.00 | 1.58 |
| RCM [14] | 11.98 | 9.29 | 14.23 | 6.97 | 4.89 | 3.89 | 10.60 | 7.84 | 11.68 | 6.67 | 3.67 | 3.14 |
| SMNA [15] | 9.07 | 8.15 | 11.28 | 6.44 | 4.54 | 3.61 | 9.23 | 5.80 | 8.39 | 4.53 | 3.10 | 2.39 |
| FAST-MATTN [7] | 45.28 | 14.40 | 28.20 | 7.19 | 7.84 | 4.67 | 39.05 | 19.88 | 30.63 | 11.6 | 11.28 | 6.08 |
| SIA [16] | 41.53 | 31.53 | **44.67** | 16.28 | **22.41** | 11.56 | 48.61 | **30.80** | **44.56** | 14.85 | **19.02** | 9.20 |
| RecBERT [11] | 16.78 | 30.67 | 35.02 | 24.90 | 18.77 | 15.27 | 15.86 | 29.61 | 32.91 | 23.99 | 16.50 | **13.51** |
| HAMT | 14.08 | **32.95** | 36.84 | **30.20** | 18.92 | **17.28** | 13.62 | 30.40 | 33.41 | **26.67** | 14.88 | 13.08 |

SR and nDTW respectively than the second place. Nevertheless, there is still a large gap compared to the human performance. We further present results on val seen and val unseen splits in Table 4.

## C.3 REVERIE dataset

The remote object localization task in REVERIE dataset requires both navigation and object grounding. To support the two subtasks in HAMT, we concatenate object features with original view image features for each viewpoint, and add an object grounding head to predict the target object given output embeddings of all object tokens. We fine-tune HAMT that is end-to-end pretrained on R2R dataset, and use the optimized ViT to extract object features given groundtruth object bounding boxes in REVERIE dataset. As shown in Table 5, HAMT achieves better navigation performance (SR and SPL), but the object grounding performance (RGS and RGSPL) on test split is worse than state of the art. Since HAMT can more effectively encode observed visual scenes and actions in the history sequence, it is able to better understand house environments and navigate to target viewpoints more efficiently as shown in the much higher SPL score. However, as we use ViT optimized on R2R dataset to extract object features, the object representation might not be as generalizable as object features used in previous works which are pretrained on large-scale object detection datasets.

## D Additional ablations

### D.1 History in training with proxy tasks

We show that the history input plays a critical role for training with proxy tasks. We compare HAMT with history input and PREVALENT [17] without history. For fair comparison, we re-implement PREVALENT which only takes instruction $\mathcal{W}$ and single-step observation $\mathcal{O}_t$ as input and the other architectures are set the same as HAMT. We train PREVALENT with all proxy tasks except the ITM task because there is no trajectory input in PREVALENT for instruction-trajectory matching. ViT features pretrained on ImageNet are used in this experiment.

In Figure 2, we present the single-step action prediction (SAP) accuracy of HAMT and PREVALENT during the training. The SAP accuracies on val seen split are similar

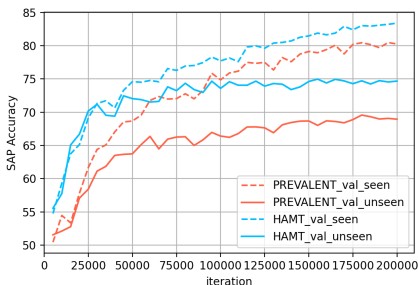

Figure 2: SAP accuracy of PREVA-LENT (w/o history) and HAMT (w/ history) on R2R dataset.

for the two models, however, PREVALENT performs much worse on the val unseen split than HAMT. Due to the capacity of large-scale transformer, PREVALENT is likely to memorize the map structure of seen houses, and thus achieves comparable performance to HAMT. However, such knowledge cannot be transferred to unseen houses because the structure and visual observations are distinct for seen and unseen houses. Feeding history as inputs avoids the model simply cramming the structure of seen houses, and enables it to align the history with an instruction to predict actions for better generalization. After fine-tuning the two models on R2R dataset, we obtain SPL 57.5 on val unseen split for HAMT, while 52.7 for PREVALENT without history input. As the same proxy tasks are used in training, the large gains of our HAMT model contribute to the history encoding. Therefore, **the proposed history encoding can largely improve the navigation performance on top of training proxy tasks.**

## D.2 Visual features in training with proxy tasks

Table 6 provides an additional experiment in the third row compared to Table 3a in main paper. It demonstrates that ViT features outperform ResNet152 features with and without training proxy tasks. Comparing the last two rows in Table 6, end-to-end feature optimization improves SPL by 2.1% on val unseen split but decreases SPL by 0.8% on val seen split. Note that we follow previous VLN works [11, 18] to select the best model based on val unseen and use the same model for val seen split. We observe that the performance on val seen split can be improved with longer

Table 6: Comparison of features (same notations as Table 3a in main paper).

| Features | PT | e2e | Val Seen | | Val Unseen | |
| --- | --- | --- | --- | --- | --- | --- |
| | | | SR | SPL | SR | SPL |
| Resnet | × | × | 65.5 | 61.3 | 54.4 | 48.7 |
| 152 | ✓ | × | 69.3 | 64.8 | 63.5 | 57.5 |
| | × | × | 68.8 | 66.1 | 56.3 | 52.5 |
| ViT | ✓ | × | **75.7** | **72.5** | 64.4 | 58.8 |
| | ✓ | ✓ | 75.0 | 71.7 | **65.7** | **60.9** |

training time. After optimizing visual representations, HAMT converges faster on val unseen split and achieves the best performance at earlier iterations. Therefore, the performance on val seen split is slightly worse than no end-to-end optimization. If training longer, the performance with optimized ViT features on val seen split can be higher.

## D.3 Different proxy tasks in end-to-end training

Table 7: Comparison of different proxy tasks in end-to-end optimization.

| SAP(R) | SPREL | Val Seen | | Val Unseen | |
| --- | --- | --- | --- | --- | --- |
| | | SR | SPL | SR | SPL |
| × | × | 70.1 | 65.9 | 63.3 | 57.7 |
| ✓ | × | 72.5 | 69.2 | 64.5 | 59.4 |
| ✓ | ✓ | **75.0** | **71.7** | **65.7** | **60.9** |

In Table 3b of main paper, we fix ViT features to ablate contributions of different proxy tasks in training. We further present the ablation results in a fully end-to-end training setup in Table 7, where different proxy tasks are used to train HAMT including the ViT features. The results show the same trend as Table 3b in main paper, where our proposed two new proxy tasks (SAP/R and SPREL) are beneficial. Moreover, we can see that the end-to-end ViT features are superior to fixed ViT features on val unseen split for all the three proxy task combinations.

## D.4 Two-stage end-to-end (e2e) training strategy

We compare our two-stage e2e training strategy with a single-stage e2e training of HAMT. However, single-stage e2e training achieves inferior performance to the two-stage training or even no e2e training. When trained for 25k iterations and evaluated on the val unseen split, the single-stage e2e training of HAMT results in SPL 53.5 while no e2e training achieves SPL 56.5. We hypothesize that the single-stage e2e training is less effective for VLN given (a) the limited training data available for the VLN task and (b) the higher complexity of VLN compared to common vision and language tasks.

## D.5 History encoding in long-horizon VLN task

We compare different history encoding approaches on the R2R-Back dataset to show that the history information is more beneficial for the long-horizon VLN task. Table 8 presents navigation results. All the models are initialized from weights after training with proxy tasks. In order to successfully return back, the agent should remember the way it comes to the targets. The recurrent state is insufficient to capture all the information and achieves the worst navigation performance. Encoding agent's oriented view at each step in temporal-only model improves over the recurrent approach. However, as the

oriented view of the agent in backward trajectory is different from the view in forward trajectory, temporal-only model does not take advantage of the full memory in previous exploration and performs inferior to our hierarchical history encoding model. It demonstrates the effectiveness of our proposed method in long-horizon VLN task that requires long-term dependency. We also show that using the end-to-end trained ViT features further benefits the navigation performance.

Table 8: Navigation results for R2R-Back dataset.

| History Encoding | e2e | Val Seen | | | | | Val Unseen | | | | |
|---|---|---|---|---|---|---|---|---|---|---|---|
| | | TL | SR↑ | SPL↑ | nDTW↑ | SDTW↑ | TL | SR↑ | SPL↑ | nDTW↑ | SDTW↑ |
| Recurrent | × | 22.33 | 51.4 | 48.4 | 67.3 | 45.7 | 23.35 | 41.1 | 37.7 | 58.2 | 35.6 |
| Temporal-only | × | 22.70 | 51.6 | 49.6 | 67.8 | 46.7 | 22.93 | 45.1 | 42.9 | 62.7 | 40.2 |
| Hierarchical | × | 23.52 | **66.8** | **63.5** | **73.8** | **60.4** | 24.58 | 56.5 | 51.7 | 63.6 | 48.4 |
| Hierarchical | ✓ | 22.76 | 64.8 | 61.8 | 73.7 | 58.9 | 23.78 | **57.2** | **53.1** | **65.1** | **49.5** |

## D.6 Structure variants in fine-tuning

Our model reuses the $f_{\text{SAP}}(o'_i \odot x'_{\text{cls}})$ in training proxy tasks to sequentially predict action in fine-tuning. In Table 9, we compare using different input tokens for the action prediction in $f_{\text{SAP}}$, including different combinations of the observation token $o'_i$, global history token $h'_{\text{cls}}$ and special text token $x'_{\text{cls}}$. We can see that the performance varies little on the val unseen split, which indicates that the cross-modal transformer in our model is able to effectively fuse different modalities so that the performance is influenced little by tokens used in prediction.

Table 9: Comparison of using different tokens in $f_{\text{SAP}}$ in fine-tuning.

| Action Prediction Token | | | Val Seen | | Val Unseen | |
|---|---|---|---|---|---|---|
| obs | txt | hist | SR↑ | SPL↑ | SR↑ | SPL↑ |
| ✓ | × | × | 76.1 | 72.8 | **66.0** | 60.3 |
| ✓ | ✓ | × | 75.0 | 71.7 | 65.7 | **60.9** |
| ✓ | × | ✓ | **78.0** | **75.9** | 65.5 | 60.2 |
| ✓ | ✓ | ✓ | 76.3 | 73.4 | 65.5 | 60.9 |

# E Qualitative results

Figures 3-6 illustrate trajectories obtained by our HAMT model and compare them to results of the state-of-the-art RecBERT [11] model. We can see that HAMT enables to better interpret instructions (Figure 3), recognize the scene (Figure 4), follow the correct direction (Figure 5), and align the current observation with the instruction (Figure 6). We also provide some failure cases in Figures 7-8, where the HAMT model still needs improvements on scene and object recognition.

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

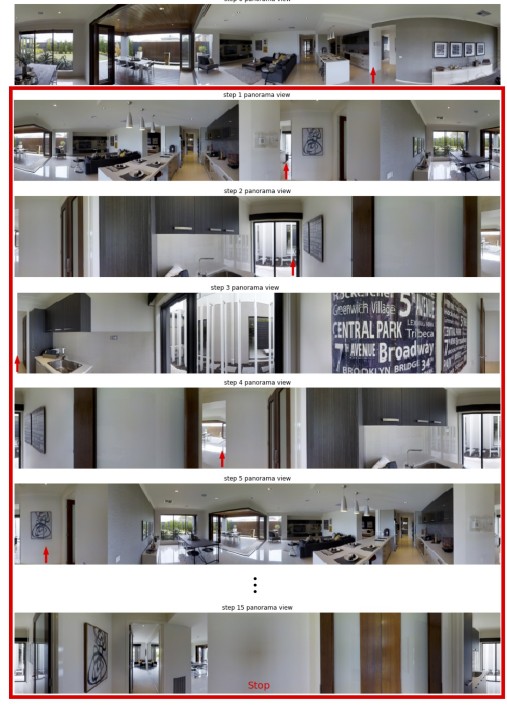

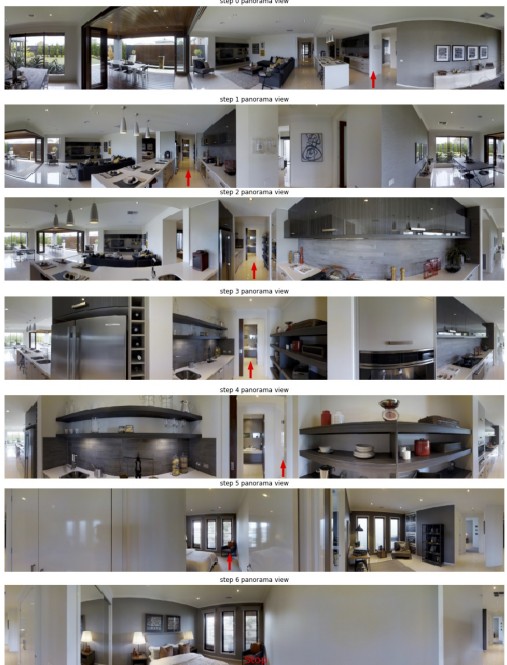

(a) Predicted trajectory by RecBERT [11] (failed).      (b) Predicted trajectory by HAMT (succeed).

Figure 4: Examples in R2R val unseen split. Navigation steps inside red box are incorrect. The instruction is "Walk into the kitchen area. Walk by the sink and oven. Walk straight into the hallway. Turn right into the little room. Turn left and walk into the bedroom. Stop by the corner of the bed." (id: 155_0). The RecBERT fails to recognize the kitchen area and navigates back and forth in wrong locations. Our HAMT correctly recognizes the kitchen and follows the instruction.

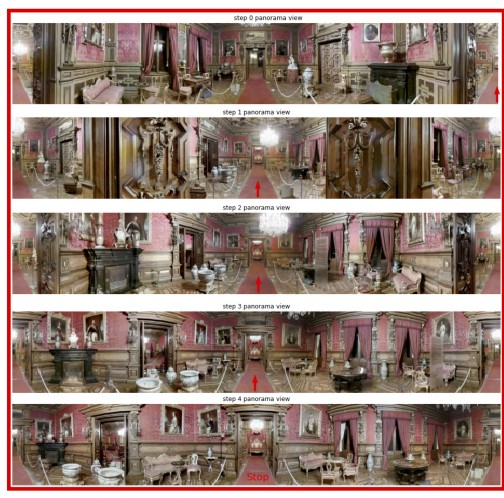

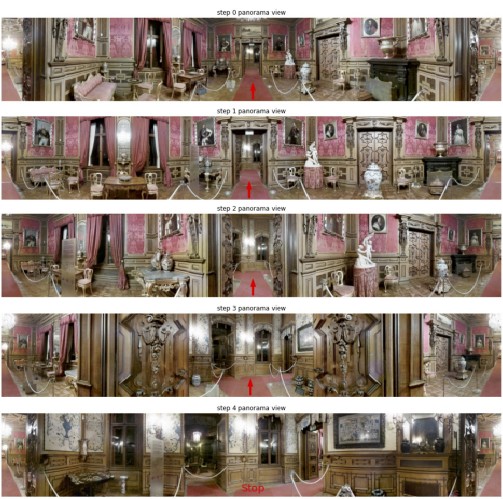

(a) Predicted trajectory by RecBERT [11] (failed).      (b) Predicted trajectory by HAMT (succeed).

Figure 5: Examples in R2R val unseen split. Navigation steps inside red box are incorrect. The instruction is "Walk straight until you get to a room that has a black table on the left with flowers on it. Wait there." (id: 4182_2). The RecBERT takes the wrong direction at the first step, while our HAMT follows the instruction and successfully stops.

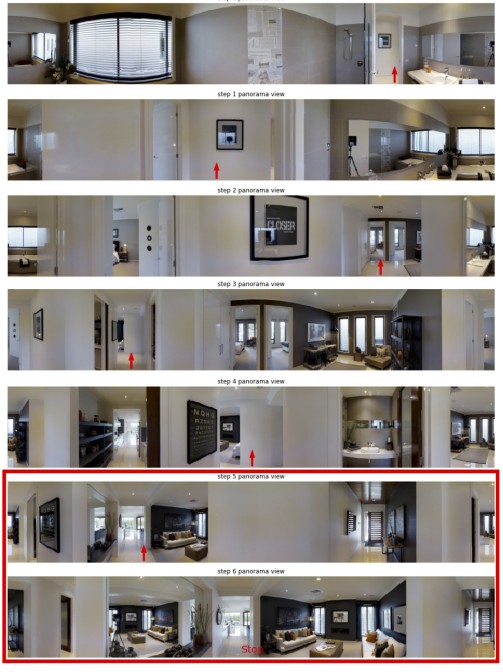

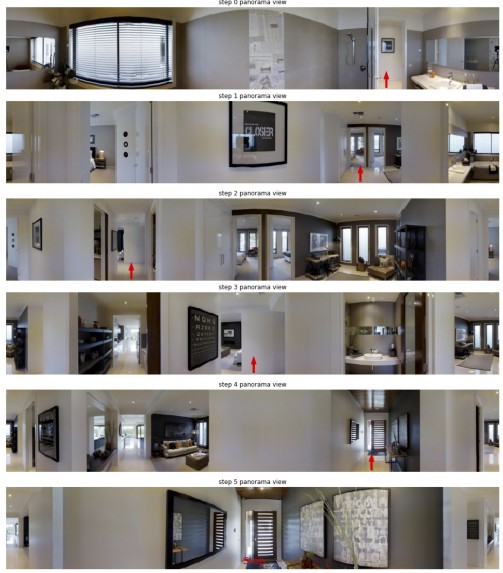

(a) Predicted trajectory by RecBERT [11] (failed).  (b) Predicted trajectory by HAMT (succeed).

Figure 6: Examples in R2R val unseen split. Navigation steps inside red box are incorrect. The instruction is "Walk out of the bathroom and turn right. Turn left and walk down the hallway. Turn right and stop by the end table." (id: 5153_0). The RecBERT correctly performs the first two turns but fails to track the third turn right action and stops incorrectly. Our HAMT is better to align the current state with the instruction to correctly perform the third turn right action.

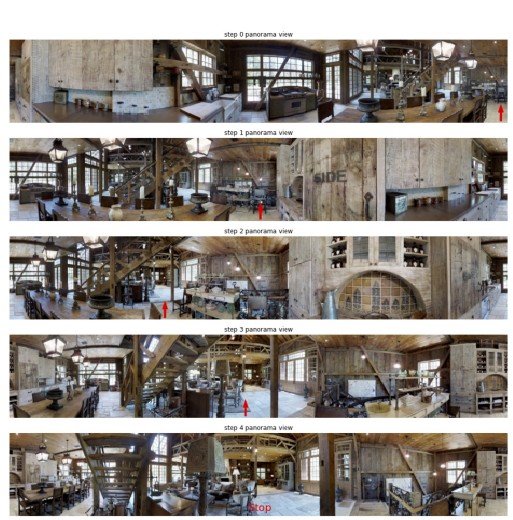

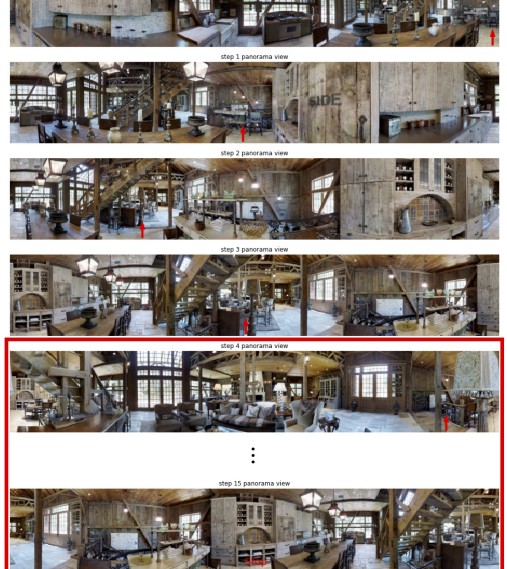

(a) groundtruth trajectory.  (b) Predicted trajectory by HAMT (failed).

Figure 7: Failure cases in R2R val unseen split. The instruction is "Go stand underneath the stairs, next to the liquor shelf. " (id: 36968_2). Though HAMT correctly goes towards the direction, it fails to recognize the liquor shelf and results in exploring further the room until reaching the maximum number of navigation steps.

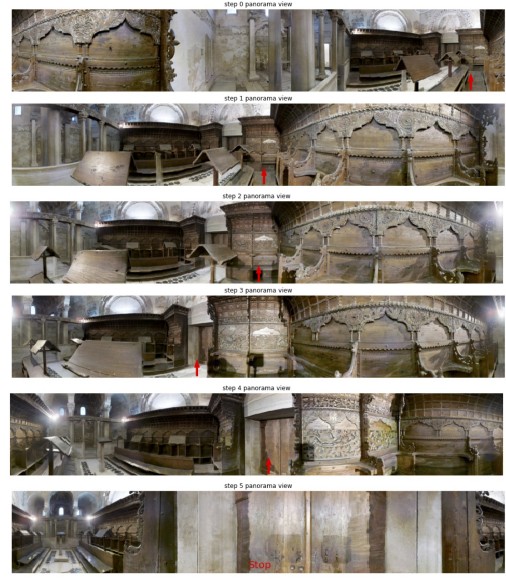 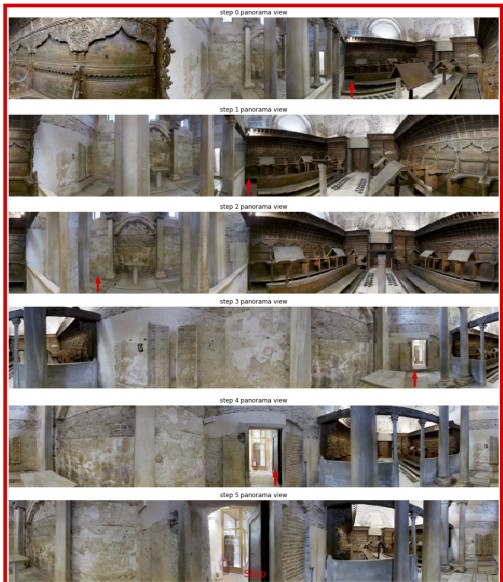

(a) Groundtruth trajectory.          (b) Predicted trajectory by HAMT (failed).

Figure 8: Failure cases in R2R val unseen split. The instruction is "With the low stone or concrete barrier behind you, walk parallel to the board covering the floor and turn left before reaching the end. Move forward to leave the wooden flooring and when on the stone flooring, turn right and stand in front of the doors leading out of the room." (id: 5873_1). As the scene is unusual, HAMT fails to locate itself in the correct direction at the first step.

[13] Alec Radford, Jong Wook Kim, Chris Hallacy, Aditya Ramesh, Gabriel Goh, Sandhini Agarwal, Girish Sastry, Amanda Askell, Pamela Mishkin, Jack Clark, et al. Learning transferable visual models from natural language supervision. In *Proceedings of the 38th International Conference on Machine Learning*, pages 8748–8763, 2021. 3

[14] Xin Wang, Qiuyuan Huang, Asli Celikyilmaz, Jianfeng Gao, Dinghan Shen, Yuan-Fang Wang, William Yang Wang, and Lei Zhang. Reinforced cross-modal matching and self-supervised imitation learning for vision-language navigation. In *Proceedings of the IEEE/CVF Conference on Computer Vision and Pattern Recognition*, pages 6629–6638, 2019. 4

[15] Chih-Yao Ma, Jiasen Lu, Zuxuan Wu, Ghassan AlRegib, Zsolt Kira, Richard Socher, and Caiming Xiong. Self-monitoring navigation agent via auxiliary progress estimation. In *Proceedings of the International Conference on Learning Representations*, 2019. 4

[16] Xiangru Lin, Guanbin Li, and Yizhou Yu. Scene-intuitive agent for remote embodied visual grounding. In *Proceedings of the IEEE/CVF Conference on Computer Vision and Pattern Recognition*, pages 7036–7045, 2021. 4

[17] Weituo Hao, Chunyuan Li, Xiujun Li, Lawrence Carin, and Jianfeng Gao. Towards learning a generic agent for vision-and-language navigation via pre-training. In *Proceedings of the IEEE/CVF Conference on Computer Vision and Pattern Recognition*, pages 13137–13146, 2020. 4

[18] Hao Tan, Licheng Yu, and Mohit Bansal. Learning to navigate unseen environments: Back translation with environmental dropout. In *Proceedings of the 2019 Conference of the North American Chapter of the Association for Computational Linguistics: Human Language Technologies, Volume 1 (Long and Short Papers)*, pages 2610–2621, 2019. 5