# OpenReview forum: "History Aware Multimodal Transformer for Vision-and-Language Navigation"
_NeurIPS.cc/2021/Conference — NeurIPS 2021 Poster_

### Official Review · Reviewer_7aBY · 2021-07-16

**Rating:** 4
**Confidence:** 5

**Summary:**

This paper presents a transformer-based model for VLN, which makes use of history trajectory information. Panoramic visual observations of all passed viewpoints are embedded by two transformer modules, one handling panorama at each step and the other one handling panorama representations of all steps. The well-known pre-training technology is employed. Experiments on R4R, R2R, and R2R variants are conducted.

**Limitations And Societal Impact:**

Thanks for the response from the authors. Most of my issues have been addressed, such as more results on downstream tasks and clarifications about model details. Overall, I think this is a good work but not good enough for NeurIPS. This is because most of the performance comes from the application of existing pre-training instead of the proposed history-aware scheme (10% vs 2% improvement), and there is no novelty in neither pretraining model design nor loss functions.

I have read initial comments from other reviewers, which tend to be positive. However, if this paper does not get lucky due to my review, I suggest rewriting the paper in line with the contribution importance of each proposed module, and paying extra attention to the newly designed xxx-back VLN task which usually can be a separate article.

**Main Review:**

The main contribution of this work are twofold: (1) introducing navigated trajectory into transformers in the form of hierarchical fashion; (2) employing the pre-training technology. Experiments are conducted on two commonly used datasets R2R and R4R. The authors also created two additional datasets, R2R-Back and R2R-Last, to further demonstrate the effectiveness of the proposed history-aware model.

The writing is not very clear as some crucial details are missing. For example, (1) Do ViT and Panoramic Transformer are shared between time steps as shown in Figure 2a? (2) Is the panorama at t=0 fed into the model for steps after t=0? If so, is the repeated calculation necessary? (3) Regarding observation encoding, 36 view embeddings are added with navigable embeddings as shown in Figure 1. However, one view may have zero or multiple navigable viewpoints. How are such cases handled by the proposed model? (4) How do you guarantee and verify the information is not lost regarding the claim in line 156 "In this way, xxx without information loss".

Regarding the performance, the main contribution hierarchical history module only brings 1%~2% SPL improvement on unseen split (which is the main reference) as shown in Table 2 (row Recurrent vs row Hierarchical), but the pre-triaing technology brings about 10% improvement. Performance is further improved if ViT is used. However, the pre-training technology is not the contribution of this work as they are taken from existing work. Thus, the claimed contribution is not matched with the experimental results. Furthermore, some ablation studies are missing, such as for the ViT setting in Table 3a, how about remove both PT and e2e? What are the contributions of SAP and SAR  respectively to the final performance?  In addition, it is weird not using the final model (ViT+PT+e2e) to do the ablation study in Table 3b.

For comparisons on R2R-Back and R2R-Last, it is not very fair for existing methods as they are not designed for such tasks. In addition, do you retrain them on R2R-Back and R2R-Last datasets?

Overall, I think this work is not ready for publication based on the novelty, writing, technique contribution, and experiment issues listed above and below.

Other issues:
1. Why zero out v_*^o or a_*^o (line 198) for spatial relationship prediction? How does it affect the performance if not zero out them?
2. Is there any explanation for the performance drop when optimizing visual representation in ViT setting of Table 3a?
3. The explanation for marginal SPL improvement in line 328 is not reasonable, because for the R2R task the SPL metric is a good measurement for path fidelity as the ground-truth path is exactly the shortest path.
4. Since pre-training technology is utilised, the proposed model should be tested on more downstream tasks, such as Touchdown, REVERIE, CVDN, etc.

**Time Spent Reviewing:**

5

---

> ### Author Response · Authors · 2021-08-10
> **Response to Reviewer 7aBY**
>
> We thank the reviewer for detailed and constructive comments. We address the raised points in the following.
>
> **Q1: Details about the method.**
>
> **(1) Do ViT and Panoramic Transformer are shared between time steps as shown in Figure 2a?**
>
> Yes, the ViT and Panoramic Transformer share parameters for different timesteps. We will add a clarification on this to the final paper version.
>
> **(2) Is the panorama at t=0 fed into the model for steps after t=0? If so, is the repeated calculation necessary?**
>
> The panorama embedding at step t=0 will be attended for action prediction at all the following steps t>0 to capture long-range dependencies across time. We empirically show that encoding the long histories in HAMT achieves better performance compared to recurrent models in Table 2. To avoid repeated computations, the panorama embeddings can be computed only once at each new timestamp and saved in memory for future processing. We demonstrate that the inference time of HAMT is only 1.1-1.5 times slower compared to a similar model without history encoding (see time comparison with RecBERT [5] in response to Q2 of Reviewer xpbr).
>
> **(3) Regarding observation encoding, 36 view embeddings are added with navigable embeddings as shown in Figure 1. However, one view may have zero or multiple navigable viewpoints. How are such cases handled by the proposed model?**
>
> During pre-training we use exactly 36 views. Following [17] we set the view as navigable if it has one or multiple navigable viewpoints.
>
> During fine-tuning, to differentiate navigable viewpoints inside one view, we follow [5, 22] and provide different view embeddings for these navigable viewpoints. Such view embeddings share the same visual features but differ by angle features, hence, more than 36 views might be provided. We will clarify details in the final paper version.
>
> **(4) How do you guarantee and verify the information is not lost regarding the claim in line 156 "In this way, xxx without information loss".**
>
> Our model keeps and attends to all panorama encoding observed during previous timestamps. In contrast, recurrent methods [5] compress the history into a single vector and are prone to losing critical information. The “temporal-only” variant of HAMT (see Figure 2(c)) keeps encodings for oriented views only and is prone to losing contextual information. Our ablation studies on R2R (Table 2) and R2R-Back (Table 2 in supplementary material) empirically demonstrate the superiority of the hierarchical history encoding. We will make our claim regarding information loss more accurate in the revised paper version.
>
> **Q2: Regarding the performance, the main contribution hierarchical history module only brings 1%~2% SPL improvement in Table 2, but the pre-training technology brings about 10% improvement. Performance is further improved if ViT is used. However, the pre-training technology is not the contribution of this work as they are taken from existing work. Thus, the claimed contribution is not matched with the experimental results.**
>
> We would like to emphasize that 1.7% SOTA improvement brought by our method is considered non-trivial for the R2R benchmark. For example, recent methods [13, 14, 27] all report improvements in the order of 1-2%. Moreover, our hierarchical history module provides +14% improvement on the more challenging R2R-Back dataset with longer trajectories, where more extended temporal reasoning is required (see Table 7).
>
> To disentangle the contributions of pretraining and history encoding on our final results, we have performed additional experiments and compared HAMT to a similar model without history encoding. To this end, we have pretrained a PREVALENT-like architecture [17] using the same proxy tasks as for HAMT (except ITM, not applicable) and used it to initialize the Recurrent baseline (see Section 4.2). The resulting model achieves SPL 52.3 on R2R val unseen vs. 57.5 for HAMT as reported in Table 3(a). This demonstrates that the history encoding adds a significant contribution on top pretraining.
>
> In addition, our pretraining method is different from previous works [17, 44] as presented in Table 1. We propose two new proxy tasks and demonstrate their effectiveness in Table 3(b). The HAMT is also the only model that is directly useable after pretraining as shown in Table 4, which reduces the gap between pretraining and fine-tuning.
>
> **Q3: More ablation studies**
>
> **(1) The ViT setting in Table 3a, how about remove both PT and e2e**
>
> We report results of HAMT when removing PT and e2e in the ViT setting. From the table below we can draw the same conclusions as from Table 3(a) in the paper: 1) the ViT features outperform ResNet152; 2) pretraining and end-to-end feature optimization are beneficial.
>
> |  |  |  | Val Seen |  | Val Unseen |  |
> |---|---|---|---|---|---|---|
> | Feature | PT | e2e | SR | SPL | SR | SPL |
> | ViT | x | x | 68.8 | 66.1 | 56.3 | 52.5 |
> | RN152 | x | x | 65.5 | 61.3 | 54.4 | 48.7 |
> | ViT | &radic; | &radic; | 75.0 | 71.7 | 65.7 | 60.9 |
>
> **(2) The contributions of SAP and SAR respectively to the final performance**
>
> The ablation results for SAP and SAR are presented in the following table. Compared to results in Table 3(b). SAP benefits more on the val seen split, while SAR is better for generalization on the val unseen split. The combination of SAP and SAR provides a balanced performance on both seen and unseen splits.
>
> |  |  | Val Seen |  | Val Unseen |  |
> |---|---|---|---|---|---|
> | SAP | SAR | SR | SPL | SR | SPL |
> | &radic; | x | 74.1 | 71.1 | 63.5 | 58.0 |
> | x | &radic; | 71.5 | 68.3 | 64.8 | 59.6 |
> | &radic; | &radic; | 75.7 | 72.5 | 64.4 | 58.8 |
>
>
> **(3) It is weird not using the final model (ViT+PT+e2e) to do the ablation study in Table 3b.**
>
> As we mention on L.264-267, e2e training requires more resources and additional training time. Therefore, we use ViT+PT to carry out the ablation study for different proxy tasks and then train with the best setting end-to-end. The ablations using ViT+PT+e2e are in the table below. Notably, these new experiments demonstrate the same trend as in Table 3(b).
>
> |  |  | Val Seen |  | Val Unseen |  |
> |---|---|---|---|---|---|
> | SAP(R) | SPREL | SR | SPL | SR | SPL |
> | x | x | 70.1 | 65.9 | 63.3 | 57.7 |
> | &radic; | x | 72.5 | 69.2 | 64.5 | 59.4 |
> | &radic; | &radic; | 75.0 | 71.7 | 65.7 | 60.9 |
>
>
> **Q4: For comparisons on R2R-Back and R2R-Last, it is not very fair for existing methods as they are not designed for such tasks. In addition, do you retrain them on R2R-Back and R2R-Last datasets?**
>
> As mentioned in Table 7 and 8 (footnote on Page 9), we use the released codes of EnvDrop and RecBERT and re-train these models on the R2R-Back and R2R-Last datasets. Since R2R-Back and R2R-Last are more challenging variations of the original R2R task, we believe that EnvDrop and RecBERT are the most appropriate baselines for our new tasks. Our new tasks require more sophisticated temporal reasoning and highlight the benefits of the history encoding in our model. At the same time, we observe that existing methods, and RNN methods in particular, have limited performance when faced with longer trajectories.
>
> **Q5: Other issues**
>
> **(1) Why zero out v_\*^o or a_\*^o (line 198) for spatial relationship prediction?**
>
> Because both visual and angle features alone are helpful to recognize the spatial relationship (see also L.196-197).
>
> **(2) Is there any explanation for the performance drop when optimizing visual representation in ViT setting of Table 3a?**
>
> The e2e improves SPL by 2.1% on the val unseen split but decreases SPL by 0.8% on the val seen split. Note that we follow previous work on VLN and select the best model based on the val unseen split. The same model is then used for the val seen split. We observe that the performance on val seen can be improved with longer training time. After optimizing the visual representations, HAMT converges faster on val unseen split and achieves the best performance at earlier iterations. Therefore, the performance on val seen is slightly worse than no e2e training. If training longer, the performance of HAMT e2e on val seen split is higher.
>
> **(3) The explanation for marginal SPL improvement in line 328 is not reasonable, because for the R2R task the SPL metric is a good measurement for path fidelity as the ground-truth path is exactly the shortest path.**
>
> Line 328 is to explain why RL improves more on SR metric than on SPL metric. In RL training, the agent can explore a viewpoint that is not in the ground-truth path. In that case, the shortest path from that viewpoint to the destination does not necessarily follow the instruction. As the reward mainly considers the shortest path instead of path fidelity with instructions, the improvements on SPL are smaller than SR.

---

> > ### Author Response · Authors · 2021-08-10
> > **Response to Reviewer 7aBY (2)**
> >
> > **(4) Since pre-training technology is utilised, the proposed model should be tested on more downstream tasks, such as Touchdown, REVERIE, CVDN, etc.**
> >
> > We have evaluated our model on R2R, R4R, R2R-Back and R2R-Last datasets and consider the superior performance on these datasets is a good indicator validating our contributions.
> >
> > To further show the generalization ability of HAMT, we evaluated HAMT on three additional datasets: REVERIE, CVDN and RxR. As shown in the following tables, HAMT achieved SOTA navigation performance on all these datasets.
> >
> > - REVERIE dataset: contains high-level instructions and requires object grounding.
> >
> > |  | Val Unseen |  |  |  | Test Unseen |  |  |  |
> > |---|---|---|---|---|---|---|---|---|
> > |  | SR | SPL | RGS | RGSPL | SR | SPL | RGS | RGSPL |
> > | SIM | 31.53 | 16.28 | 22.41 | 11.56 | 30.80 | 14.85 | 19.02 | 9.20 |
> > | RecBERT (init. OSCAR) | 25.53 | 21.06 | 14.20 | 12.00 | 24.62 | 19.48 | 12.65 | 10.00 |
> > | RecBERT (init. PREVALENT) | 30.67 | 24.90 | 18.77 | 15.27 | 29.61 | 23.99 | 16.50 | 13.51 |
> > | HAMT (ours) | 32.95 | 30.20 | 18.92 | 17.28 | 30.40 | 26.67 | 14.88 | 13.08 |
> >
> > We achieved better navigation performance (SR, SPL), but the object grounding performance (RGS, RGSPL) on the test set is worse than SOTA. To be noted, we use the e2e ViT to extract object features, which might not be as generalizable as the pretrained BUTD object features used in RecBERT.
> >
> > - CVDN dataset: contains dialogs as instruction and longer navigation steps than R2R.
> >
> > |  | Goal Progress (m) |  |  |
> > |---|:---:|:---:|:---:|
> > |  | Val Seen | Val Unseen | Test Unseen |
> > | PREVALENT | - | 3.15 | 2.44 |
> > | VISITRON | 5.11 | 3.25 | 3.11 |
> > | MT-RCM+EnvAg | 5.07 | 4.65 | 3.91 |
> > | HAMT (ours) | 6.91 | 5.13 | 5.58 |
> >
> > - RxR dataset on testing leaderboard: contains more longer instructions and navigation steps than R2R. ([https://ai.google.com/research/rxr/competition?active_tab=leaderboard](https://ai.google.com/research/rxr/competition?active_tab=leaderboard) )
> >
> > | Model | SR | SPL | NDTW | SDTW |
> > |---|---|---|---|---|
> > | Monolingual Baseline | 25.40 | 22.59 | 41.05 | 20.59 |
> > | CLIP-ViL | 38.34 | 35.17 | 51.10 | 32.42 |
> > | CLEAR-CLIP | 38.45 | 34.50 | 51.72 | 32.96 |
> > | HAMT (ours) | 53.12 | 46.62 | 59.94 | 45.19 |

---

> ### Author Response · Authors · 2021-09-01
> **Response to Additional Comments from Reviewer 7aBY**
>
> We thank the reviewer for additional comments.
>
> To disentangle the contributions of pretraining and history encoding, we carried out additional experiments (see Q2 for more details in our original rebuttal).  In a nutshell, when using the same pretraining, we observe SPL performance of 52.3 for the model *without* history encoding (Recurrent) and 57.5 for our model *with* history encoding (HAMT) on the R2R dataset. This demonstrates a **5.2% improvement** of our model over an equivalent model without history encoding on R2R.
>
> The improvement is even higher on the R2R-Back dataset: HAMT achieves 53.1 and the SoTA recurrent model 37.7 (both models are pretrained on the same data). This is a **15.4% improvement** over the SoTA Recurrent model. The larger gain can be explained by the fact that the R2R-Back task requires longer history reasoning and hence the proposed history-aware architecture brings more significant gains.

---

### Official Review · Reviewer_euGp · 2021-07-18

**Rating:** 7
**Confidence:** 5

**Summary:**

This paper introduces a History Aware Multimodal Transformer to incorporate a long-horizon history into multimodal decision making. The authors perform experiments on three VLN datasets (R2R, R4R, and R2R-last), and demonstrate that HAMT is particularly effective for navigation tasks with longer trajectories.


**Limitations And Societal Impact:**

Yes.

**Main Review:**

This paper introduces a History Aware Multimodal Transformer to incorporate a long-horizon history into multimodal decision making. The authors perform experiments on three VLN datasets (R2R, R4R, and R2R-last), and demonstrate that HAMT is particularly effective for navigation tasks with longer trajectories.

Strengths:
- A new multimodal transformer based method for VLN is introduced particularly for solving the longer paths.
- The proposed HAMT outperforms the state of the art and achieves excellent results on three VLN datasets.
- Training HAMT with auxiliary proxy tasks is interesting.

Weaknesses:
- More clarification and ablation study is needed about the training strategy. Have the authors tried to train HAMT in a single stage end-to-end rather than the current two-stage strategy?
- Experimental details were not made clear and many important hyper-parameters are missing, making this paper hard to reproduce. Are all the Panoramic Transformer layers randomly initialized, or are they initialized from certain checkpoints? How do you initialize the discount factor? Besides, the authors introduce a new VLN framework, thus it is important to report some model details and training details, for example, the numbers of model parameters, memory cost and inference time (the model seems to be rather complex), and so on.
- Line 160, how do you get the special token? From my guess you are treating it as a totally free parameter during training. If so, how do you initialize it?
- The presentation of the paper can be further improved. For example, the authors could give a brief formulation of Temporal Transformer, Panoramic Transformer, and Multimodal Transformer first, and then give a unified and specific description.

**Time Spent Reviewing:**

3

---

> ### Author Response · Authors · 2021-08-10
> **Response to Reviewer euGp**
>
> We thank the reviewer for acknowledging our contributions and for constructive comments to our work.
>
> **Q1: Have the authors tried to train HAMT in a single stage end-to-end rather than the current two-stage strategy?**
>
> Yes, we have tried the single-stage end-to-end (e2e) training of HAMT. Results of such training, however, are inferior to the two-stage training and no e2e training. For example, when trained for 25k iterations and evaluated on the val unseen split, the single-stage e2e training of HAMT results in SPL 53.5 while no e2e training achieves SPL 56.5.  We hypothesize that the single-stage e2e training is less effective for VLN given (a) the limited training data available for the VLN task and (b) the higher complexity of VLN compared to other common vision and language tasks [42]. We will report these results and add a discussion on one-stage vs. two-stage training in the final version of the paper, if accepted.
>
> **Q2: Experimental details were not made clear and many important hyper-parameters are missing, making this paper hard to reproduce.**
>
> We provide additional details about our method below and will include them in the final version of the paper. Implementation details including all hyper-parameters of our method are also available from our source code in the supplementary material. To make our work reproducible, we will release our code and pre-trained models upon acceptance.
>
> **(1) Are all the Panoramic Transformer layers randomly initialized, or are they initialized from certain checkpoints?**
>
> The panoramic transformer layers are randomly initialized.
>
> **(2) How do you initialize the discount factor?**
>
> We follow previous works [5, 22] and set the discount factor to 0.9.
>
> **(3) It is important to report some model details and training details, for example, the numbers of model parameters, memory cost and inference time, and so on.**
>
> Our HAMT model contains ~170M parameters. It consumes ~9G of GPU memory with batch size of 8 at inference, and the inference time is ~44ms per sample in the val unseen split. The HAMT variant without Text-to-Vision attention (see Table 4 of supplementary material) contains ~170m parameters, takes ~5G of GPU memory with batch size of 8 at inference and the inference time is ~32ms per sample on the val unseen split. (Please refer to the details of evaluation settings in the reply Q2 for Reviewer xpbr).
>
> **Q3: Line 160, how do you get the special token? From my guess you are treating it as a totally free parameter during training. If so, how do you initialize it?**
>
> Yes, the special token is a parameter to learn. We initialize it by the zero vector. We will add details to the final paper.
>
> **Q4: The presentation of the paper can be further improved. For example, the authors could give a brief formulation of Temporal Transformer, Panoramic Transformer, and Multimodal Transformer first, and then give a unified and specific description.**
>
> Thank you for your suggestion. We will add corresponding formulations to the overview of HAMT in Section 3.1.

---

### Official Review · Reviewer_E9jz · 2021-07-19

**Rating:** 8
**Confidence:** 5

**Summary:**

This paper proposes a history-aware multimodal transformer (HAMT) style model for Vision-and-Language Navigation task which explicitly takes the history of observations as the input for the transformer model. HAMT efficiently encodes all past panoramic observations using a hierarchical vision transformer. It consists of unimodal transformers for text, history (of panoramic visual observations), and current visual observation and then fuses these representations together using a cross-modal transformer. They also propose additional auxiliary proxy tasks for end-to-end training of HAMT. These are Single-step Action Prediction & Regression (SAP/SAR) and Spatial Relationship Prediction (SPREL). Through extensive ablations, they show that how each of their contributions makes sense and gets state-of-the-art on the single-run R2R task, R4R, and R2R-Back & R2R-Last (two augmented tasks they described in the paper).

**Main Review:**

### Originality and Significance:
-	This paper proposes a novel modelling approach where they explicitly model the history of panoramic visual observations in addition to instruction and current visual observation. Their “history-aware” transformer style model first encodes individual images with Vision Transformer and then combine it with unimodal transformer output of instruction and current visual observation using a cross-modal transformer.
-	They also propose 2 new auxiliary tasks for pretraining their model: Single-step Action Prediction/Regression and Spatial Relationship Prediction.
-	They show results on 2 VLN tasks: R2R and R4R.
    -	Using ablations, they show that their history encoding for VLN improves Success Rate by 4.2% and 2.1% SPL over temporal-only baseline.
    -	They show that using all their proxy tasks and visual representation finetuning provides gains of 12.2% SPL (48.7 -> 60.9) and 11.3% SR (54.4 -> 65.7).
    -	They achieve state-of-the-art on R2R and beats recurrent-VLN-BERT by 3% on SPL and 2% on SR.
    -	They get state-of-the-art on R4R and two additional VLN setup tasks (introduced in the paper), R2R-Back where the agent needs to remember the trajectory and come back to starting position, R2R-Last where only the last instruction is provided.

### Quality and Clarity:
The paper is well-written, is clear about their claims and explanations and is easy to follow.

### Typos:
L98: and-to-end -> end-to-end


**Time Spent Reviewing:**

10

---

> ### Author Response · Authors · 2021-08-10
> **Response to Reviewer E9jz**
>
> We thank the reviewer for acknowledging the originality and significance of our work. We will correct typos in the final version of the paper, if accepted.

---

### Official Review · Reviewer_xpbr · 2021-07-19

**Rating:** 6
**Confidence:** 4

**Summary:**

This paper proposed history-aware multi-modal transformers for vision and language navigation tasks. Different from prior work (RecBert, PREVALENT), the proposed method takes long-horizon history with vision transformers and spatial relationship prediction proxy tasks. The proposed method achieves the state of the art performance on VLN R2R tasks.

**Limitations And Societal Impact:**

Yes, the author addresses the limitation and potential negative social impact of the work.

**Main Review:**

[Strength]

- Very good performance on VLN R2R datasets.

- The idea of encoding the whole history as input is intuitive, and the paper is well written and the authors perform extensive ablation study to verify the effectiveness of each module.

- The author also try with different visual backbone and imitation learning and A3C for the tasks.

[Weakness]

- In the contribution section, the author claims their model is the first fully transformer-based model for VLN tasks. This is not super clear to me, it will be better to illustrate more precisely.

- Since the model encodes the whole history image as input, I wonder what is the computation time during inference time.

**Time Spent Reviewing:**

3 hours

---

> ### Author Response · Authors · 2021-08-10
> **Response to Reviewer xpbr**
>
> We thank the reviewer for providing constructive comments and for acknowledging thorough experiments and the superior performance of our model.
>
> **Q1: In the contribution section, the author claims their model is the first fully transformer-based model for VLN tasks. This is not super clear to me, it will be better to illustrate more precisely.**
>
> Our HAMT model uses transformers at all stages, i.e. for language encoding, visual encoding and multimodal decision making as shown in Figure 1 and 2(a). In contrast, other recent VLN methods [6, 11, 22, 26, 27, 28, 29] mostly adopt precomputed ResNet features for visual encoding and combine them with LSTM for language encoding and decision making. A few recent works exploit transformers for language encoding [5, 15, 17, 32] or multimodal decision making [5, 17, 32]. However, in contrast to our work, none of the previous methods explore transformers for visual encoding. Moreover, we fine-tune transformer-based visual representations for the VLN task and show how this strategy results in significant improvements beyond state of the art, see Table 3(a) and Table 5. Given these results, we believe the exploration of a fully transformer-based model together with its end-to-end training for VLN tasks makes a significant contribution to our work.
>
> **Q2: Since the model encodes the whole history image as input, I wonder what is the computation time during inference time.**
>
> To assess the influence of history encoding on the inference time, we compare our model to RecBERT [5]. Our HAMT and RecBERT use the same number of layers in the language transformer and cross-modal transformer. The main difference of two models is in the history encoding and the attended length of history for action prediction. We run each model on the R2R validation unseen split (2349 instructions) and report inference times averaged over two runs using a single Tesla P100 GPU. For our method we compare variants with and without Text-to-Vision Attention, denoted here as HAMT and HAMT noT2V respectively (see Section 3.3 of the main paper and Section C.3 in supplementary material for more details).
>
> | Method | Inference Time | SR | SPL |
> |:---:|:---:|:---:|:---:|
> | RecBERT [5] | 69s | 63 | 57 |
> | HAMT | 104s | 66 | 61 |
> | HAMT noT2V | 76s | 65 | 60 |
>
> We can see that HAMT and its noT2V variant are only 1.5x and 1.1x slower compared to RecBERT, suggesting that attending to the whole history does not increase the inference time significantly. Moreover, while HAMT noT2V is only 10% slower compared to [5], it still outperforms SOTA in SP and SPL on Val Unseen.

---

### Author Response · Authors · 2021-08-10
**Generic Comment to All Reviewers**

We thank all the reviewers for their helpful comments and insightful suggestions. In this rebuttal, we mainly address the following points raised by the reviewers:
- the memory footprint and inference time of our model;
- additional details on the method;
- additional ablation studies for our model;
- SOTA performance on three additional datasets (REVERIE, CVDN and RxR).

---

### Decision · Program_Chairs · 2021-09-27

**Decision:**

Accept (Poster)

**Comment:**

This paper presents an effective history-aware multi-modal transformer for VLN tasks, which captures well the long-horizon history and spatial relationships.  This method efficiently encodes all past panoramic observations using a hierarchical vision transformer, with promising performances on a variety of datasets.  It achieves SOTA on R2R.  The main concerns raised by reviewers are the unclear writing and the relatively smaller gap from using a new history-aware scheme compared to pretraining models. The authors later provide extensive experiments to demonstrate the generalization capability and large improvements by their modules in more downstream tasks and more diverse datasets.  The experiments indeed validating well the contribution of the proposed transformers. Considering the positive comments by three reviewers and the responses solve the most issues by reviewer 7aBY, the AC thus recommends accepting this paper.